# Auditory fear memory retrieval requires BLA-LS and LS-VMH circuitries via GABAergic and dopaminergic neurons

Miao Chen [1,2], Jun Li[1], Weiran Shan [1], Jianjun Yang[1,2] & Zhiyi Zuo [1✉]

## Abstract

**Fear and associated learning and memory are critical for developing defensive behavior. Excessive fear and anxiety are important components of post-traumatic stress disorder. However, the neurobiology of fear conditioning, especially tone-related fear memory retrieval, has not been clearly defined, which limits specific intervention development for patients with excessive fear and anxiety. Here, we show that auditory fear memory retrieval stimuli activate multiple brain regions including the lateral septum (LS). Inhibition of the LS and the connection between basolateral amygdala (BLA) and LS or between LS and ventromedial nucleus of the hypothalamus (VMH) attenuates tone-related fear conditioning and memory retrieval. Inhibiting GABAergic neurons or dopaminergic neurons in the LS also attenuates tone-related fear conditioning. Our data further show that fear conditioning is inhibited by blocking orexin B signaling in the LS. Our results indicate that the neural circuitries BLA–LS and LS-VMH are critical for tone-related fear conditioning and memory retrieval, and that GABAergic neurons, dopaminergic neurons and orexin signaling in the LS participate in this auditory fear conditioning.**

**Keywords** Fear Conditioning; Fear Memory Retrieval; GABAergic Neurons; Lateral Septum; Orexin
**Subject Category** Neuroscience

## Introduction

Fear and associated learning and memory induce defensive behavior to avoid danger or harm, which is a basic survival behavior and is preserved in many animal species, including humans (Bolles, 1970; Tovote et al, 2015). However, excessive fear and the associated anxiety are huge burdens on the affected individuals and important components of post-traumatic stress disorder (Hamner et al, 1999; Saggu et al, 2023; Tovote et al, 2015). Thus, understanding the mechanisms for the development of fear and the associated learning and memory has a great implication.

The most commonly used paradigm to examine fear and the associated learning and memory in animal studies is fear conditioning test. The learning and memory with this test can be divided into hippocampus-dependent (context-related) and hippocampus-independent (tone-related) (Kim et al, 1992). Various brain regions including hippocampus, amygdala, and prefrontal cortex have been shown to be involved in context-related fear conditioning (Grosso et al, 2018; Kim et al, 1992; Kim et al, 2006). Glutamate receptor-dependent plasticity is critical for this learning and memory (Kim et al, 2006; Sah et al, 2003). The amygdala is known to be involved in tone-related fear conditioning (Kim et al, 1992, 2006). However, tone-related fear conditioning, which usually induces a higher fear behavior (freezing behavior) than that induced by context-related fear conditioning (Lin et al, 2021), is understudied.

Lateral septum (LS) plays a critical role in motivated behaviors and receives input from the hippocampus (Kim et al, 2006; Rizzi-Wise et al, 2021). LS is involved in the development of context-related fear conditioning (Reis et al, 2010; Yeates et al, 2022). LS is activated by auditory fear stimuli, including auditory fear memory retrieval stimuli (Butler et al, 2015; Garcia et al, 1996; Holschneider et al, 2006). Infusing lidocaine to LS during the application of conditioning stimuli attenuates tone-related fear conditioning (Calandreau et al, 2007). Infusion of agents modulating glutamate neurotransmission into the LS immediately before conditioning stimuli regulates tone-related fear conditioning (Calandreau et al, 2010). However, it is not clear whether LS plays a role in tone-related fear memory retrieval. This study was designed to determine whether the neural circuitries/brain regions, such as LS, were involved in the development of tone-related fear conditioning and memory retrieval. The molecules for this involvement were also investigated. Our results suggest that the basolateral amygdala (BLA)-LS and LS-ventromedial nucleus of the hypothalamus (VMH) neural circuitries are critical for tone-related fear conditioning and memory retrieval. GABAergic and dopaminergic neurons that are activated by orexin B in the LS play a role in this fear conditioning.

[1]Department of Anesthesiology, University of Virginia, Charlottesville, VA 22908, USA. [2]Department of Anesthesiology, Pain and Perioperative Medicine, The First Affiliated Hospital, Zhengzhou University, Zhengzhou, Henan, China. ✉E-mail: zz3c@virginia.edu

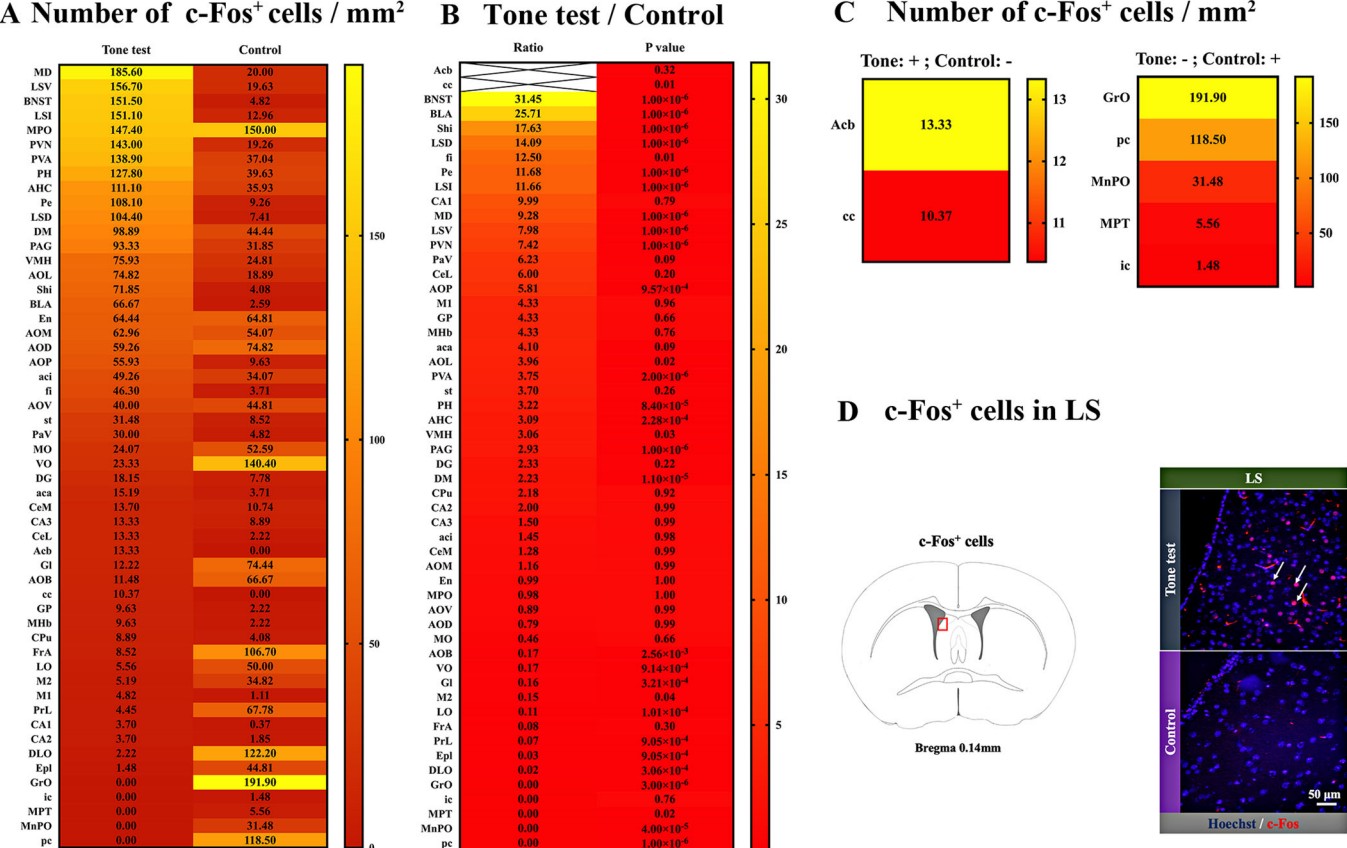

**Figure 1. Tone-dependent fear conditioning induced changes in c-Fos expression in mouse brain.**

All mice were sacrificed for the detection of c-Fos-positive cells at 1 h after memory retrieval tone stimuli. Control mice were naive mice that were not exposed to conditioning stimuli or tone-related memory retrieval stimuli. (A) Heatmap of the number of c-Fos-positive cells in brain regions of mice with tone-related fear conditioning stimuli and control mice (n = 5 mice in each group). (B) Heatmap of the comparisons in brain regions between mice with tone-related fear conditioning stimuli and control mice (n = 5 mice in each group, statistical analysis was done by t test). (C) Heatmap showing special cases of c-Fos-positive cell patterns. Left panel: no c-Fos-positive cells in control mice. Right panel: no c-Fos-positive cells in mice with tone-related fear conditioning. −: no c-Fos positive cells. +: with c-Fos-positive cells. (D) Representative images of c-Fos immunohistochemistry in the LS. The red rectangle in the left panel indicates the location of the photomicrographs. The white arrows indicate c-Fos-positive cells. aca anterior commissure, anterior part, Acb nucleus accumbens aci anterior commissure, intrabulbar part, AHC anterior hypothalamic area, central part, AOB accessory olfactory bulb, AOD anterior olfactory area, dorsal part, AOL anterior olfactory area, lateral part, AOM anterior olfactory area, medial part, AOP anterior olfactory area, posterior part, AOV anterior olfactory area, ventral part, BLA basolateral amygdaloid nucleus, BNST bed nucleus of stria terminalis, CA1 cornu ammonis region 1 of the hippocampus, CA2 cornu ammonis region 1 of the hippocampus, CA3 cornu ammonis region 1 of the hippocampus, cc corpus callosum, CeL lateral central amygdala nucleus, CeM medial central amygdala nucleus, CPu caudate putamen (striatum), DG dentate gyrus, DLO dorsolateral orbital cortex, DM dorsomedial hypothalamic nucleus, En endopiriform claustrum, Epl external plexiform layer of the olfactory bulb, fi fimbria of the hippocampus, FrA frontal association cortex, Gl glomerular layer of the olfactory bulb, GP globus pallidus, GrO granule cell layer of the olfactory bulb, ic internal capsule, LO lateral orbital cortex, LSD lateral septal nucleus, dorsal part, LSI lateral septal nucleus, intermediate part, LSV lateral septal nucleus, ventral part, M1 primary motor cortex, M2 secondary motor cortex, MD mediodorsal thalamic nucleus, MHb medial habenular nucleus, MnPO median preoptic nucleus, MO medial orbital cortex, MPO median preoptic nucleus, MPT medial pretectal nucleus, PAG periaqueductal grey, PaV paraventricular hypothalamic nucleus, ventral part, pc paracentral thalamic nucleus, Pe periventricular hypothalamic nucleus, PH post hypothalamic area, PrL prelimbic cortex, PVA paraventricular thalamic nucleus, PVN paraventricular hypothalamic nucleus, Shi septohippocampal nucleus, st stria terminals, VMH ventromedial hypothalamic nucleus, VO ventral orbital cortex. Source data are available online for this figure.

# Results

## Tone-related fear conditioning activated multiple brain regions including LS

Mouse brain was harvested 1 h after the completion of the three tone stimuli (memory retrieval stimuli) that occurred 24 h after the four paired foot shock and tone stimuli (training) in the fear conditioning (Appendix Fig. S1A). Multiple brain regions, including bed nucleus of stria terminalis, BLA and LS, had increased c-Fos expression after the tone stimuli (Fig. 1; Appendix Fig. S1A), suggesting that tone-related memory retrieval stimuli

activate these brain regions. We selected LS as our focus because LS is involved in motivated behavior and anxiety and has been shown to contribute to the development of context-related fear conditioning (Reis et al, 2010; Rizzi-Wise et al, 2021; Yeates et al, 2022). However, its role in tone-related fear memory retrieval has not been defined yet.

## GABAergic and dopaminergic neurons in the LS were involved in tone-related fear conditioning

To determine whether LS is involved in tone-related fear conditioning, mice received the injection of virus carrying the

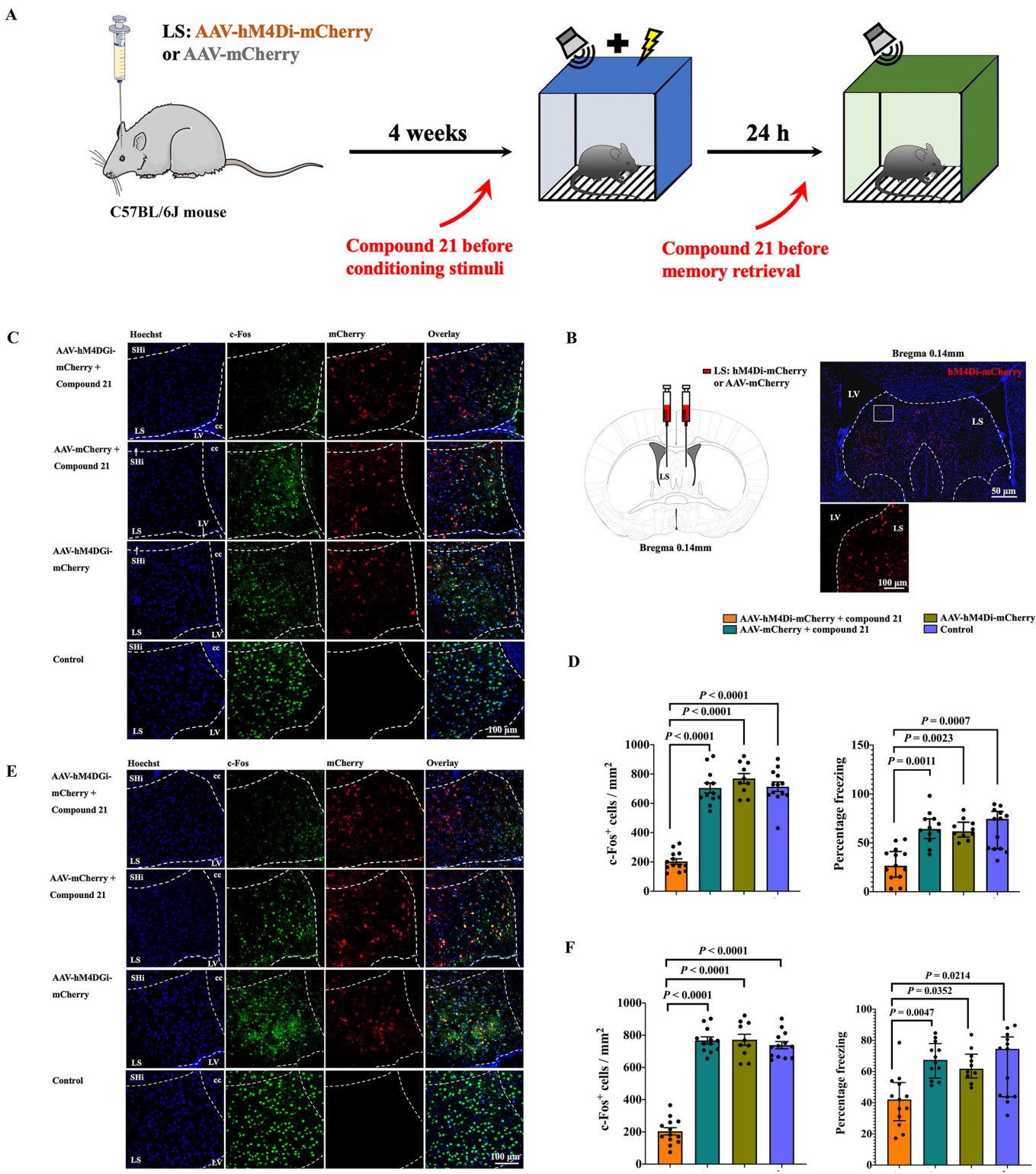

code of hM4Di into the LS 4 weeks before they were subjected to fear conditioning. Mice received compound 21 30 min before the application of the training stimuli or memory retrieval stimuli (Fig. 2A,B). The combination of hM4Di and compound 21 reduced the number of cells expressing c-Fos in the LS no matter whether

the compound 21 was injected 30 min before the training stimuli or memory retrieval stimuli. These injections also decreased tone-related fear conditioning (freezing behavior). However, the injection of the control virus plus compound 21 or virus carrying the code of hM4Di alone did not affect the expression of c-Fos and

**Figure 2.  LS was essential for learning and memory retrieval of fear conditioning.**

(A) A schematic of the experimental design. Compound 21 was injected intraperitoneally 30 min before either conditioning stimuli or memory retrieval stimuli. Mice were sacrificed for the detection of c-Fos-positive cells at 1 h after memory retrieval stimuli. (B) A Schematic of the chemogenetic approach to inhibit the LS neurons and representative photomicrographs showing the expression of mCherry in the LS of mice with injection. The white rectangle indicates the location of the bottom micrograph. The dashed white lines indicate the boundary of the LS. (C, D) Compound 21 was given before conditioning tone stimuli. (C) Representative images of c-Fos immunohistochemistry in the LS. The dashed lines indicate the boundary of the LS. SHi septohippocampal nucleus, cc corpus callosum, LV lateral ventricle. (D) Left panel: quantification of c-Fos expression [$n = 10$–13 mice per group; $F_{(3,38)} = 80.17$, $P < 0.0001$; Brown–Forsythe and Welch one-way analysis of variance test]. Right panel: performance during fear conditioning [$n = 10$–13 mice per group; $H = 21.426$ (df $= 3$) on rank, $P < 0.0001$; Kruskal–Wallis test]. Data are presented as mean ± SEM. (normal distribution data) or median ± interquartile range (not normal distribution data) with the presentation of individual animal data in the bar graph. (E, F) Compound 21 was given before memory retrieval tone stimuli. (E) Representative images of c-Fos immunohistochemistry in the LS. The dashed lines indicate the boundary of the LS. (F) Left panel: quantification of c-Fos expression [$n = 10$–13 mice per group; $F_{(3,38)} = 125.3$, $P < 0.0001$; Brown–Forsythe and Welch one-way analysis of variance test]. Right panel: performance during fear conditioning [$N = 10$–13 mice per group; $H = 14.343$ (df $= 3$) on rank, $P = 0.0025$; Kruskal–Wallis test]. Data are presented as mean ± SEM (normal distribution data) or median ± interquartile range (not normal distribution data) with the presentation of individual animal data in the bar graph. Source data are available online for this figure.

tone-related fear conditioning (Fig. 2C–F). These results suggest that inhibiting neurons in the LS attenuates tone-related fear conditioning.

To determine the types of neurons that were involved in tone-related fear conditioning, brain sections were immunostained with glutamic acid decarboxylase 65 and 67 kDa isoform (GAD65&67, a GABAergic neuron marker), vesicular glutamate transporter 1 (vGluT1, a glutamatergic neuron marker) or tyrosine hydroxylase (TH, a dopaminergic neuron marker). There was no staining of vGluT1 (Appendix Fig. S2A) in the LS, suggesting that there are no glutamatergic neurons in this brain region. However, a previous study has shown that there are cells with positive staining for calcium/calmodulin-dependent protein kinase type II subunit α (Li et al, 2023), a marker for glutamatergic neurons (Mohanan et al, 2022), in the LS. These different findings regarding whether there are glutamatergic neurons in the LS may be due to the use of different markers for these neurons or anatomic locations of LS used in the staining in our study and the previous study. However, there was abundant staining of GAD65&67 and TH. Those neurons that were c-Fos-positive were also positive for GAD65&67 and TH in mice with auditory memory retrieval stimuli (Appendix Fig. S2A; Fig. 3A,B). These results suggest that the neurons activated by tone stimuli were GABAergic neurons and dopaminergic neurons. Interestingly, some cells were positive for both GABA and TH (Appendix Fig. S3A), suggesting that these neurons may use more than one neurotransmitter to affect their downstream neurons.

To determine whether GABAergic and dopaminergic neurons were involved in tone-related fear conditioning, DLAG, a glutamate decarboxylase inhibitor (Zhang et al, 2017), or L-AMPT, a tyrosine hydroxylase inhibitor (Brogden et al, 1981), was injected into the LS 3 days before fear conditioning training sessions. Glutamate decarboxylase and tyrosine hydroxylase are needed for the synthesis of GABA and dopamine, respectively. DLAG and L-AMPT reduced tone-induced c-Fos expression and freezing behavior, while the injection of vehicle solution did not affect the expression of c-Fos and freezing behavior (Fig. 3C–E; Appendix Fig. S2B). These results suggest that the GABAergic neurons and dopaminergic neurons in the LS are critical for tone-related fear conditioning.

To provide additional specific evidence for the role of GABAergic neurons and dopaminergic neurons in the LS, AAV-hSyn-DIO-hM4D(Gi)-mCherry and AAV9-mGAD2-Cre or AAV-hSyn-DIO-hM4D(Gi)-mCherry and AAV.rTH.PI.Cre.SV40 were

injected into the LS (Fig. 4A,B). Mice received the combination of these viruses 4 weeks ago and then compound 21 30 min before the fear memory retrieval stimuli had a decrease in freezing behavior and the number of c-Fos-positive cells after tone stimuli compared with control mice (Fig. 4C–E; Appendix Fig. S3B). These results suggest that GABAergic neurons and dopaminergic neurons in the LS are involved in tone-related fear memory retrieval.

## BLA–LS and LS-VMH neural circuitries were involved in tone-related fear conditioning

To determine the neural circuitry that was involved in tone-related fear conditioning, viruses for anterograde and retrograde tracing were injected into the LS (Fig. 5A). As expected, multiple brain regions sent input to the LS that sent out input to many brain regions (Fig. 5B–E; Appendix Figs. S4 and S5). Among them, we selected BLA and VMH as a brain region upstream and downstream of LS, respectively, because the amygdala is known to be involved in fear conditioning (Grosso et al, 2018; Kim et al, 1992; Kim et al, 2006), VMH plays a role in social activity (Khodai et al, 2021) and stimulating VMH induces freezing behavior (Wang et al, 2015). In addition, neurons in the BLA and VMH had increased c-Fos expression after memory retrieval tone stimuli (Fig. 1A,B). Finally, the involvement of the connections between BLA and LS or LS and VMH in fear conditioning had not been reported previously and determining this involvement would provide novel insights into neural circuitry for fear conditioning.

To investigate the role of BLA–LS and LS-VMH connections in tone-related fear conditioning, retrograde chemogenetic approach was used. Mice that received injection of AAV-DIO-hM4Di-mCherry into the LS or BLA had mCherry in the LS neurons or BLA neurons, respectively. These neurons were c-Fos staining positive (Fig. 6A–C; Appendix Figs. S6 and S7). Mice in various groups had a similar amount of freezing behavior in the cage immediately before the memory retrieval stimuli were applied (Fig. 6D), suggesting that disrupting the BLA–LS and LS-VMH connections does not affect freezing behavior in general. However, mice receiving the injection of AAV-DIO-hM4Di-mCherry into BLA and CAV2-Cre into LS 4 weeks ago and then compound 21 30 min before the fear conditioning training or auditory memory retrieval stimuli had a reduced number of c-Fos-positive cells in the BLA, LS and VMH. These mice also had a reduced tone-related freezing behavior. Mice receiving the injection of AAV-DIO-

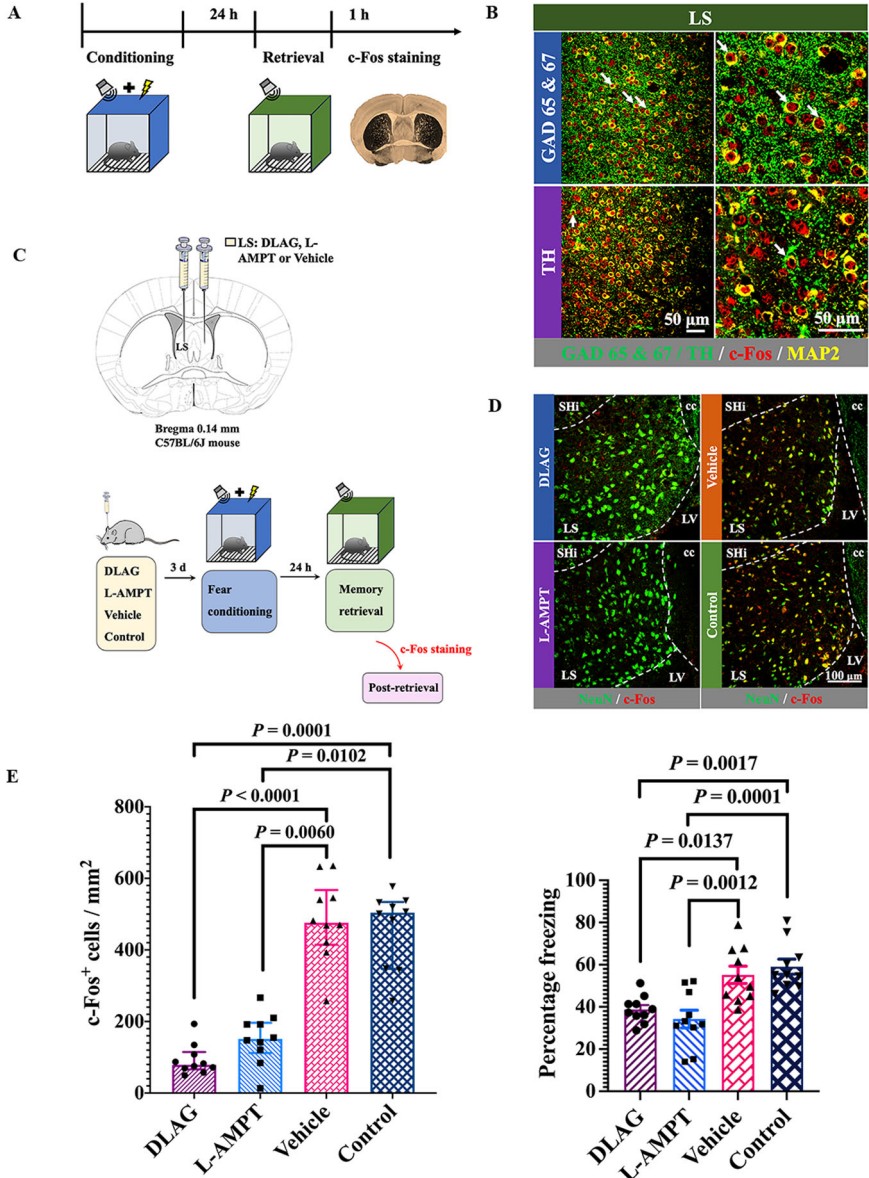

**Figure 3. GABAergic and dopaminergic neurons were critical in tone-related fear conditioning.**

(A) A schematic of the experimental design. Mice were sacrificed for the detection of c-Fos positive cells in the LS at 1 h after memory retrieval tone stimuli. (B) Representatives of immunofluorescent staining showing co-localization of GAD65&67 or TH with c-Fos in the LS, respectively. The white arrows indicate GABAergic and dopaminergic cells in the LS. GAD65&67, glutamate decarboxylase 65 & glutamate decarboxylase 67; TH, tyrosine hydroxylase; MAP2, microtubule-associated protein 2. (C) Schematic of DLAG, L-AMPT or vehicle injection and experimental timeline for examining the role of GABAergic and dopaminergic cells in the LS in fear conditioning. Mice were sacrificed for the detection of c-Fos-positive cells in the LS at 1 h after memory retrieval tone stimuli. DLAG DL-2-Allylglycine, L-AMPT α-Methyl-L-tyrosine. (D) Representatives of immunofluorescent staining showing co-localization of NeuN with c-Fos in the LS. The dashed lines indicate the boundary of the LS. SHi septohippocampal nucleus, cc corpus callosum, LV lateral ventricle, NeuN Neuronal nuclear protein. (E) Left panel: quantification of c-Fos expression [$n = 10$ mice per group; $H = 30.029$ (df $= 3$) on rank, $P < 0.0001$; Kruskal–Wallis test]. Right panel: performance during fear conditioning [$n = 10$ mice per group; $F_{(3,36)} = 11.40$, $P < 0.0001$; Brown–Forsythe and Welch one-way analysis of variance test]. Data are presented as mean ± SEM (normal distribution data) or median ± interquartile range (not normal distribution data) with the presentation of individual animal data in the bar graph. Source data are available online for this figure.

mCherry into BLA and CAV2-Cre into LS 4 weeks ago and then compound 21 30 min before the fear conditioning training or auditory memory retrieval stimuli did not have a change in c-Fos expression and freezing behavior after tone stimuli (Fig. 6E,F). These results suggest that BLA–LS pathway plays a role in tone-related fear conditioning. Similarly, mice receiving the injection of

AAV-DIO-hM4Di-mCherry into the LS and CAV2-Cre into the VMH 4 weeks ago and then compound 21 30 min before the fear conditioning training or auditory memory retrieval stimuli had a reduced number of c-Fos positive cells in the LS and VMH. These mice also had a reduced tone-related freezing behavior. Mice receiving the injection of AAV-DIO-mCherry into the LS and

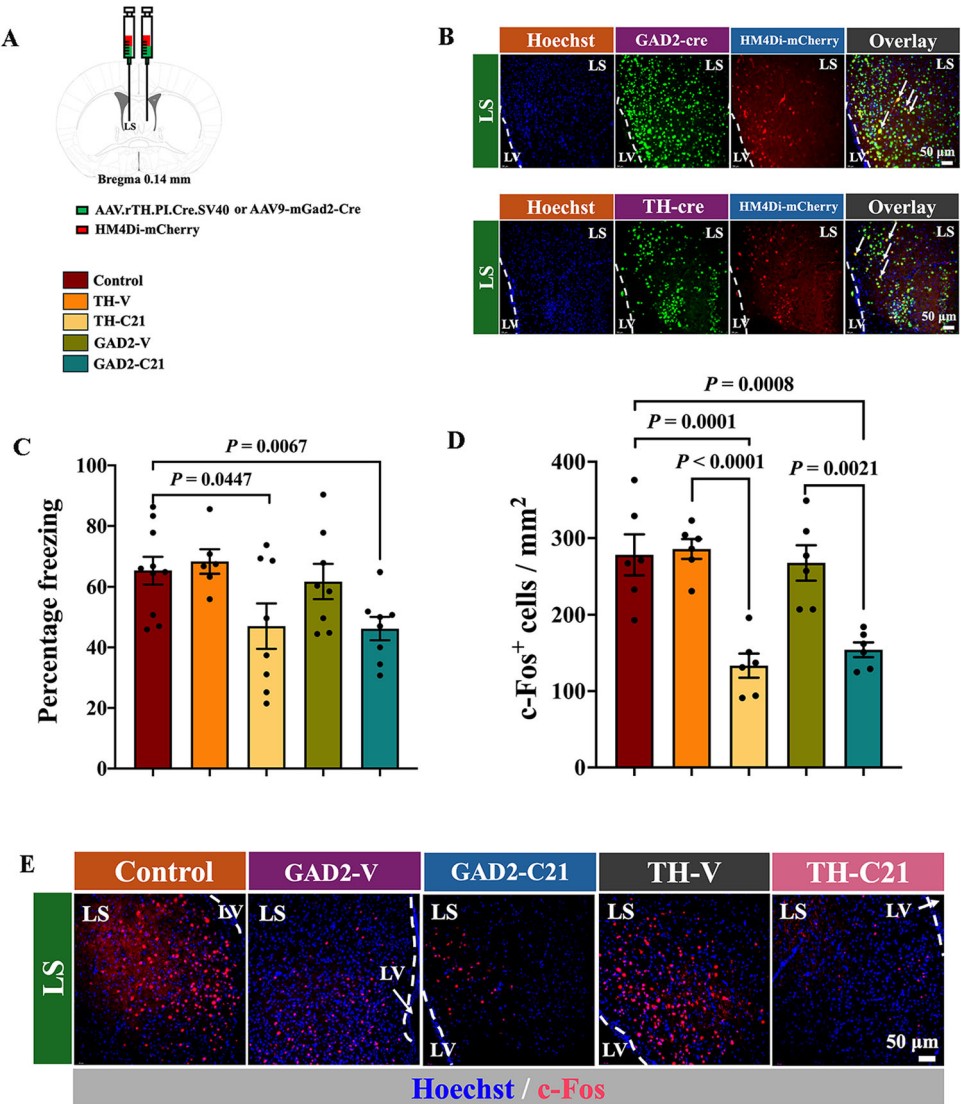

**Figure 4. GABAergic and dopaminergic neurons were critical in tone-related fear memory retrieval.**

(A) A schematic of the viral injection. Mice were used in tone-related fear conditioning test and sacrificed for the detection of c-Fos positive cells in the LS at 1 h after memory retrieval tone stimuli. (B) Representatives of immunofluorescent staining showing co-localization of Cre with mCherry (HM4Di) in the LS. The white arrows indicate co-staining cells in the LS. LV lateral ventricle. (C) Performance during tone-related memory phase [$n = 6$–10 mice per group; $F_{(4,35)} = 3.58$, $P = 0.015$; one-way analysis of variance test followed with $t$ test]. Data are presented as mean ± EM (normal distribution data) with the presentation of individual animal data in the bar graph. C21 compound 21, GAD glutamate decarboxylase, TH tyrosine hydroxylase, V vehicle. (D) Quantification of c-Fos expression [$n = 6$ mice per group; $F_{(4,25)} = 15.39$, $P < 0.0001$; one-way analysis of variance test]. Data are presented as mean ± SEM (normal distribution data) with the presentation of individual animal data in the bar graph. (E) Representatives of immunofluorescent staining showing c-Fos staining in the LS. LV lateral ventricle. Source data are available online for this figure.

CAV2-Cre into the VMH 4 weeks ago and then compound 21 30 min before the fear conditioning training or auditory memory retrieval stimuli did not have a change in c-Fos expression and freezing behavior after tone stimuli (Fig. 6G,H). These results suggest a role of LS-VMH pathway in tone-related fear conditioning. Taking together, our data indicate that the BLA–LS and LS-VMH neural circuitries are critical for tone-related fear conditioning. Interestingly, mice receiving the injection of AAV-DIO-hM4Di-mCherry into the LS and CAV2-Cre into the VMH 4 weeks ago and then compound 21 30 min before the memory retrieval stimuli had a reduced number of c-Fos staining in the BLA compared with mice that were exposed to tone stimuli but without

receiving any injections (Fig. 6G). The reason for this decrease is unclear.

To determine whether the connections between BLA and LS or LS and VMH were important specifically for the auditory fear conditioning, mice received context-related fear memory retrieval stimuli or were exposed to 2,5-dihydro-2,4,5-trimethylthiazoline (TMT), a chemical in fox feces and urine to elicit innate fear responses in rodents (Janitzky et al, 2009). Mice had increased freezing behavior after being exposed to context-related fear memory retrieval stimuli or TMT (Fig. 7A,B). The c-Fos-positive cells in the LS but not in the BLA or VMH were increased by these stimuli (Fig. 7C–H; Appendix Fig. S8). These results suggest that

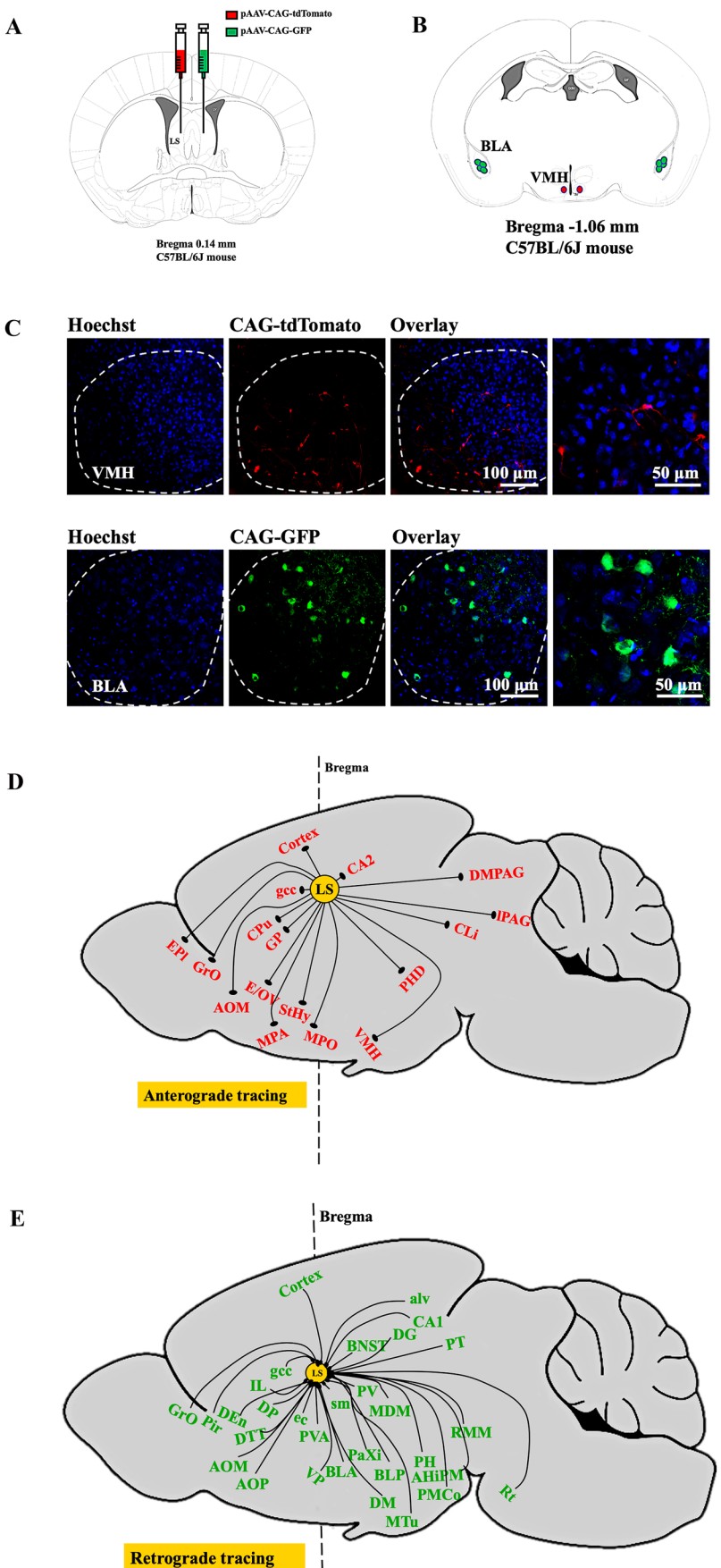

◄ **Figure 5. Anterograde and retrograde tracing from the LS by viral vectors.**

(A) A schematic of the experimental approach to anterogradely and retrogradely label neurons from and to the LS (*n* = 3 mice). pAAV-CAG-tdTomato (red) was for anterograde tracing. pAAV-CAG-GFP (green) was for retrograde tracing. (B) A schematic of VMH and BLA staining, which were positive for pAAV-CAG-tdTomato (red) and pAAV-CAG-GFP (green), respectively. VMH ventromedial hypothalamic nucleus, BLA basolateral amygdaloid nucleus. (C) Representative images of VMH and BLA labeling, which were positive for pAAV-CAG-tdTomato (red) and pAAV-CAG-GFP (green), respectively. The dashed lines indicate the boundaries of VMH and BLA. (D) A schematic of brain regions labeled by pAAV-CAG-tdTomato (red). The dashed line indicates the bregma. AOM anterior olfactory area, medial part, CA2 cornu ammonis region 2 of the hippocampus, CLi caudal linear nucleus of the raphe, CPu caudate putamen (striatum), DMPAG dorsomedial periaqueductal grey, E/OV ependymal and subendymal layer/olfactory ventricle, EPl external plexiform layer of the olfactory bulb, gcc genu corpus callosum, GP globus pallidus, GrO granule cell layer of the olfactory bulb, LPAG lateral periaqueductal grey, MPA medial preoptic area, MPO median preoptic nucleus, PHD post hypothalamic area, dorsal part, StHy striohypothalamic nucleus, VMH ventromedial hypothalamic nucleus. (E) A schematic of brain regions labeled by pAAV-CAG-GFP (green). The dashed line indicates the bregma. AA anterior amygdaloid area, AHiPM amygdalohippocampus, posteromedial area, alv alveus of the hippocampus, AOM anterior olfactory area, medial part, AOP anterior olfactory area, posterior part, BLA basolateral amygdaloid nucleus, BLP basolateral amygdala nucleus, post, CA1 field CA1 hippocampus, DEn dorsal endopiriform claustrum, DG dentate gyrus, DM dorsomedial hypothalamic nucleus, DP dorsal peduncular cortex, DTT dorsal tenia tecta, ec external capsule, gcc genu corpus callosum, GrO granule cell layer of the olfactory bulb, IL infralimbic cortex, MDM mediodorsal thalamic nucleus, medial part, MTu medial tuberal nucleus, PaXi paraxiphoid nucleus of thalamus, PH posterior hypothalamic nucleus, Pir piriform cortex, PMCo posteromedial cortical amygdala nucleus, PV paraventricular thalamic nucleus, PVA paraventricular thalamic nucleus, PT paratenial thalamic nucleus, RMM retromammillary nucleus, medial part, Rt reticular thalamic nucleus, sm stria medullaris, VP ventral pallidum. Source data are available online for this figure.

BLA or VMH is not activated by TMT or context-related fear memory retrieval stimuli. Thus, the BLA–LS and LS-VMH connections may not be important for context-related or TMP-induced fear behaviors.

## Orexin B signaling played a role in tone-related fear conditioning

To determine molecules that were involved in tone-related fear conditioning, we focused on orexin signaling because orexin receptors are expressed in the mouse LS (Tsuneoka et al, 2024) and orexin signaling is involved in regulating social behavior and arousal (Tsujino et al, 2013). The number of cells that were positively stained for orexin B but not for orexin A was increased (Fig. 8A,B; Appendix Figs. S9A,B). To determine whether orexin B played a role in tone-related fear conditioning, an anti-orexin B antibody or TCS OX2 29, a non-peptide orexin antagonist (Hirose et al, 2003), was injected into the LS (Fig. 8C). This injection decreased the number of c-Fos positive cells in the LS of mice with tone stimuli and tone-related fear conditioning behavior; whereas heat-inactivated anti-orexin B antibody and vehicle did not have an effect on the number of cells with c-Fos expression in the LS and tone-related fear conditioning behavior (Fig. 8D,E; Appendix Fig. S9C). These results suggest the role of orexin B in tone-related fear conditioning behavior.

## Discussion

Fear and associated learning and memory are critical for developing survival behavior to avoid injury. However, excessive fear-associated learning, memory and anxiety are important components of many neuropsychiatric diseases including post-traumatic stress disorder (Hamner et al, 1999; Saggu et al, 2023; Tovote et al, 2015). One of the common forms to test fear and associated learning and memory in rodents is fear conditioning. Many studies have investigated the involvement of neural circuitry in context-related fear conditioning (Grosso et al, 2018; Kim et al, 1992; Kim et al, 2006). Other than amygdala (Kim et al, 2006), the involvement of neural circuitry in tone-related fear conditioning is not clear. Our study showed that tone increased the number of

cells with c-Fos expression in the BLA, LS and VMH. Inhibiting LS neurons, the connection between BLA and LS, or the connection between LS and VMH reduced tone-related learning and memory. These results suggest that BLA–LS and LS-VMH pathways are involved in the development of tone-related learning and memory retrieval (Fig. 9). Importantly, this involvement of the pathway may be specific because disrupting this pathway did not affect the baseline freezing behavior and context-related fear memory retrieval stimuli and TMT-induced fear behavior but did not activate this pathway. Thus, the BLA–LS and LS-VMH pathways may be critical and specific for the conditioned stimulus-unconditioned stimulus connections in tone-related fear learning and memory.

The LS has been shown to be involved in reward, feeding, anxiety, fear and social behavior (Rizzi-Wise et al, 2021). Damage to the LS can induce fear responses to harmless stimuli (Brady et al, 1953), suggesting that LS inhibits fear responses. However, recent studies have shown that the LS can inhibit or increase fear conditioning (Besnard et al, 2019; Opalka et al, 2020; Rizzi-Wise et al, 2021). These previous studies are focused on determining the role of the connection/pathway between the hippocampus and LS in the context-related fear conditioning (Besnard et al, 2019; Opalka et al, 2020; Rizzi-Wise et al, 2021). Infusing lidocaine into LS during the conditioning phase inhibits tone-related freezing behavior in mice (Calandreau et al, 2007). However, whether neurons in the LS are inhibited by lidocaine is not examined in that study. Whether LS is involved in tone-related fear memory retrieval is not defined. Our results clearly indicate a role of the LS in tone-related fear conditioning. The LS is a major hub to connect the hippocampus with many subcortical regions (Rizzi-Wise et al, 2021). As anticipated, the LS sends out input to many brain regions including VMH. The LS receives input from many brain regions including the brain cortex, hippocampus and BLA (Rizzi-Wise et al, 2021). BLA is known to be involved in fear learning and social novelty recognition behavior (Rodriguez et al, 2023; Yau et al, 2021). VMH is an integral component to regulate neuroendocrine functions (Khodai et al, 2021). Studies have suggested a role of VMH in aggressive behavior (Lee et al, 2014; Lin et al, 2011). Low-frequency stimuli to VMH induce freezing behavior in rodents (Wang et al, 2015). Thus, we decided to determine the role of BLA and VMH in tone-related fear conditioning. Our anterograde and

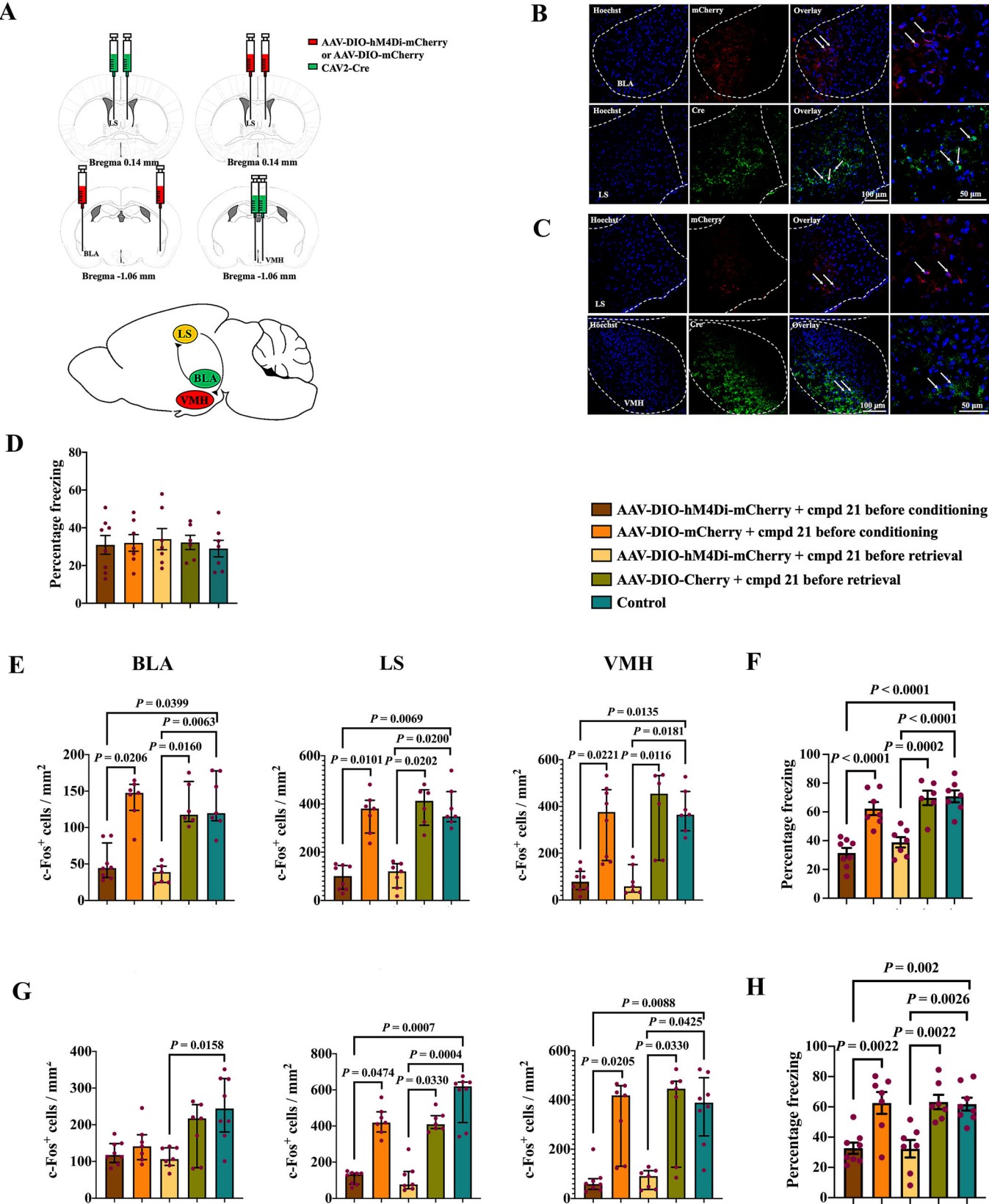

**Figure 6. Connections between BLA and LS or between LS and VMH were essential for tone-induced fear conditioning.**

(A) A schematic of the experimental approach using the AAV-DIO-hM4Di-mCherry or AAV-DIO-mCherry (red) to infect BLA neurons and CAV2-Cre (green) to target the BLA neurons that projected to LS (left two panels) or to infect LS neurons and CAV2-Cre (green) to target the LS neurons that projected to VMH (right two panels). Bottom panel: a schematic of the presumptive BLA–LS and LS-VMH neural circuitries. (B) Representative immunohistochemical staining of virus location. The white arrows indicate the neurons labeled with mCherry (red) in the BLA or nerve termini labeled with an anti-Cre recombinase antibody (green) in the LS. The dashed white lines indicate the boundaries of the BLA and LS. Right panels are high magnification images of the left panel. (C) Representative immunohistochemical staining of virus location. The white arrows indicate the neurons labeled with mCherry (red) in the LS or nerve termini labeled with an anti-Cre recombinase antibody (green) in the VMH. The dashed white lines indicate the boundaries of the LS and VMH. Right panels are high magnification images of the left panel. (D) Freezing behavior was evaluated during the 3 min before the memory retrieval tone stimuli were applied [$n = 6$–8 mice per group; $F_{(4,30)} = 0.15$, $P = 0.9615$; Brown–Forsythe and Welch one-way analysis of variance test]. Data are presented as mean ± SEM (normal distribution data) with the presentation of individual animal data in the bar graph. (E, F) Compound 21 was given to inhibit the connection between BLA and LS. Compound 21 was injected intraperitoneally 30 min before either conditioning tone stimuli or memory retrieval tone stimuli. Mice were sacrificed for the detection of c-Fos-positive cells at 1 h after memory retrieval tone stimuli. (E) Quantification of c-Fos expression in the BLA [$n = 6$–8 mice per group; $H = 24.212$ (df $= 4$) on rank, $P < 0.0001$; Kruskal–Wallis test], LS [$n = 6$–8 mice per group; $H = 25.091$ (df $= 4$) on rank, $P < 0.0001$; Kruskal–Wallis test] and VMH [$n = 6$–8 mice per group; $H = 24.194$ (df $= 4$) on rank, $P < 0.0001$; Kruskal–Wallis test]. Data are presented as median ± interquartile range (not normal distribution data) with the presentation of individual animal data in the bar graph. (F) Performance during fear conditioning [$n = 6$–8 mice per group; $F_{(4,30)} = 20.22$, $P < 0.0001$; Brown–Forsythe and Welch one-way analysis of variance test]. Data are presented as mean ± SEM (normal distribution data) with the presentation of individual animal data in the bar graph. (G, H) Compound 21 was given to inhibit the connection between LS and VMH. Compound 21 was injected intraperitoneally 30 min before either conditioning tone stimuli or memory retrieval tone stimuli. Mice were sacrificed for the detection of c-Fos-positive cells at 1 h after memory retrieval tone stimuli. (G) Quantification of c-Fos expression in BLA [$n = 7$–8 mice per group; $H = 12.117$ (df $= 4$) on rank, $P = 0.017$; Kruskal–Wallis test], LS [$n = 7$–8 mice per group; $H = 27.649$ (df $= 4$) on rank, $P < 0.0001$; Kruskal–Wallis test] and VMH [$n = 7$–8 mice per group; $H = 22.615$ (df $= 4$) on rank, $P = 0.0002$; Kruskal–Wallis test]. Data are presented as median ± interquartile range (not normal distribution data) with the presentation of individual animal data in the bar graph. (H) Performance during fear conditioning [$n = 7$–8 mice per group; $F_{(4,32)} = 10.19$, $P < 0.0001$; Brown–Forsythe and Welch one-way analysis of variance test]. Data are presented as mean ± SEM (normal distribution data) with the presentation of individual animal data in the bar graph. Source data are available online for this figure.

retrograde tracing experiment clearly showed the connections between BLA and LS and between LS and VMH. Inhibiting neurons in the BLA reduced the number of activated neurons in the LS and VMH. These results provide the anatomic and activity evidence for the existence of BLA–LS–VMH pathway. Inhibiting the connections of BLA and LS or LS and VMH attenuated tone-related fear conditioning. These results also indicate novel functions for BLA and VMH. Interestingly, the inhibition of the LS or the connection between the BLA and LS or between the LS and VMH before either the training tone stimuli or auditory memory retrieval stimuli reduced tone-related fear behavior. These results suggest that the neural circuitries BLA–LS and LS-VMH are a critical component for the auditory fear learning and memory retrieval.

The vast majority of neurons in the LS have been reported to be GABAergic neurons (Risold et al, 1997; Wong et al, 2016). Consistent with this knowledge, abundant staining of GAD65&67 was found in the LS. This staining was co-localized with c-Fos staining in tone-stimulated mice, suggesting that the neurons activated by tone are GABAergic neurons. Inhibiting these neurons by blocking the synthesis of GABA reduced tone-related fear conditioning, suggesting that GABAergic neurons in the LS are important for this fear conditioning. There was also positive staining of TH in the LS. This staining was co-localized with c-Fos staining in tone-stimulated mice, suggesting that these neurons activated by tone are dopaminergic neurons. Inhibiting these neurons by blocking the synthesis of dopamine reduced tone-related fear conditioning, suggesting that dopaminergic neurons in the LS are important for this fear conditioning. Inhibiting GABAergic or dopaminergic neurons in the LS by chemogenetic approach just before the application of tone-related fear memory retrieval stimuli reduced freezing behavior, suggesting that these GABAergic or dopaminergic neurons play a role in tone-related fear memory retrieval. Interestingly, we detected a significant amount of TH positive cells in the LS. TH positive cells have been identified previously in the LS (Sjostedt et al, 2020; The Human Protein Atlas, 2015). In addition, myeloid ecotrophic insertion

site 2, a transcription factor that facilitates dopaminergic fate specification (Agoston et al, 2014), is expressed in the mouse LS (Reid et al, 2024). Our study showed TH positive cells in the LS by immunostaining. In addition, inhibiting TH by L-AMPT or inhibiting TH-containing cells by chemogenetic approach in the LS reduced the number of activated cells in the LS after fear conditioning stimuli and attenuated the auditory fear conditioning behavior. These results suggest the existence of dopaminergic neurons in the LS.

There are many types of peptide receptors including orexin receptors in the LS neurons (Rizzi-Wise et al, 2021; Tsuneoka et al, 2024). Consistent with this knowledge, orexins existed in the LS. The number of cells having orexin B but not cells with orexin A was increased by tone stimuli. Orexin system is involved in various physiological processes including sleep–wake cycle, feeding and reward behavior, and energy balance (Inutsuka et al, 2013; Tsujino et al, 2013). Our results suggest that orexin B is involved in tone-induced neuronal activation and fear conditioning because neutralizing orexin B or antagonizing orexin reduced these effects of tone stimuli. Orexin neurons are mainly in the lateral hypothalamic area and project to many brain regions including cerebral cortex and septum (Li et al, 2023). Many brain regions including LS and amygdala innervate orexin neurons (Inutsuka et al, 2013; Tsujino et al, 2013). Our results suggest that BLA neurons regulate the activation of LS neurons via orexin signaling. Detailed mechanism on how orexin signaling mediates the regulation of LS neurons by BLA neurons needs further investigation.

Our results showed that fewer cells in the BLA were c-Fos positive when the connection between the LS and VMH was inhibited by chemogenetic approach just before memory retrieval tone stimuli were applied (Fig. 6G). It is not clear whether this difference is because the control mice that received tone stimuli without any treatment had an unexpected high level of c-Fos expressing cells (compared with results in Fig. 6E) or there is a feedback mechanism due to complex connections among brain

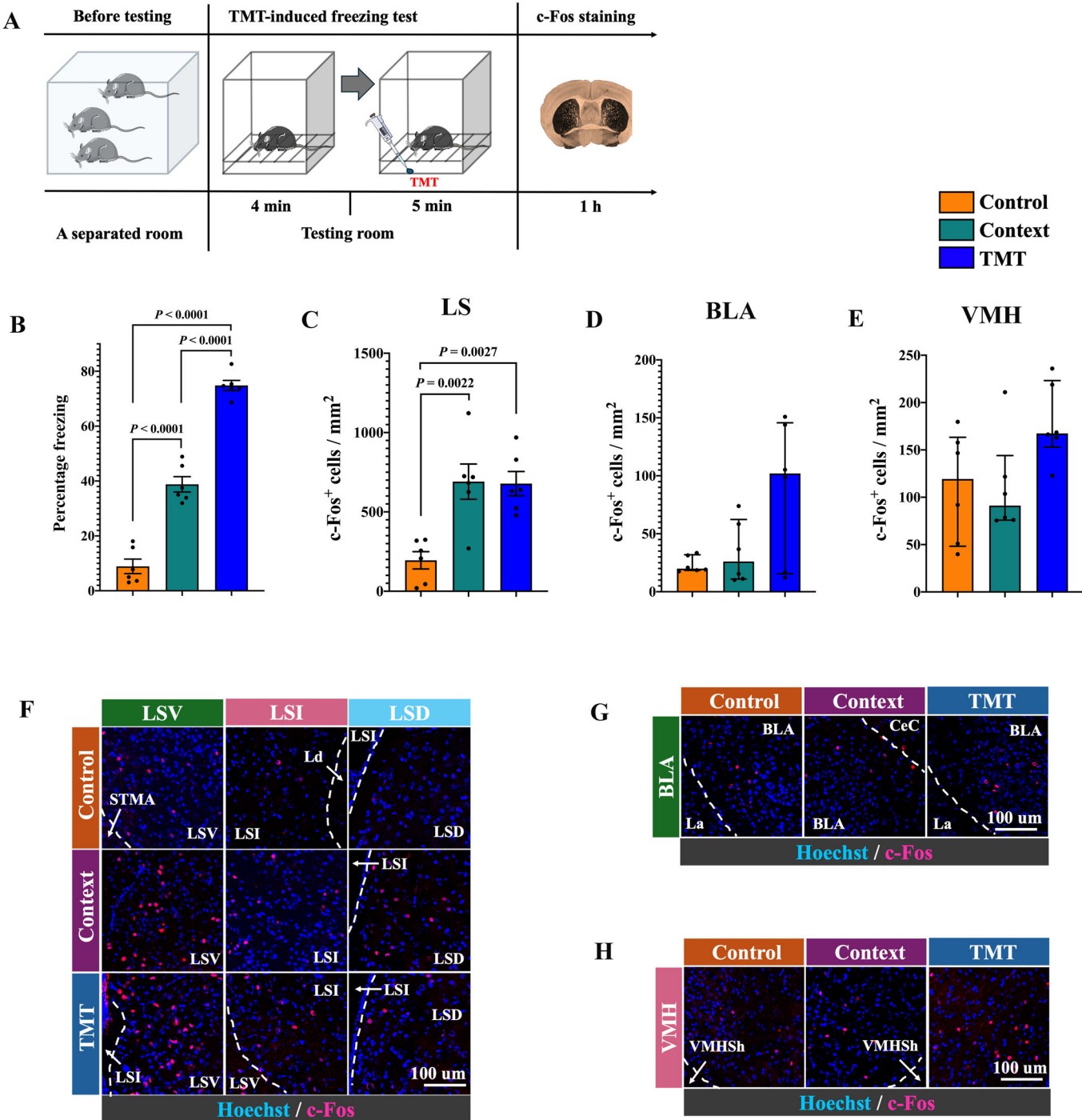

**Figure 7. Fear context or TMT-induced freezing behavior and activation of LS.**

(A) A schematic of TMT stimulation experiment. Mice were sacrificed for the detection of c-Fos-positive cells in brain regions at 1 h after being exposed to TMT. TMT, 2,5-dihydro-2,4,5-trimethylthiazoline. (B) Quantification of freezing behavior [$n = 6$ mice per group; $F_{(2,15)} = 179.4$, $P < 0.0001$; one-way analysis of variance test]. Data are presented as mean ± SEM (normal distribution data) with the presentation of individual animal data in the bar graph. (C–E) Quantification of c-Fos expression in various brain regions. (C) Quantification of c-Fos expression in the LS [$n = 6$ mice per group; $F_{(2,15)} = 11.35$, $P = 0.001$; one-way analysis of variance test]. (D) Quantification of c-Fos expression in the BLA [$n = 6$ mice per group; $H = 2.608$ (df = 2) on rank, $P = 0.2844$; Kruskal–Wallis test]. (E) Quantification of c-Fos expression in the VMH [$n = 6$ mice per group; $H = 5.942$ (df = 2) on rank, $P = 0.0452$; Kruskal–Wallis test]. Data are presented as mean ± SEM (normal distribution data) or median ± interquartile range (not normal distribution data) with the presentation of individual animal data in the bar graph. (F–H) Representatives of immunofluorescent staining showing c-Fos staining in the LS, BLA, and VMH. CeC central amygdala nucleus, capsular, La lateral amygdala nucleus, Ld lambdoid septal zone, LSD lateral septal nucleus, dorsal part, LSI lateral septal nucleus, intermediate part, LSV lateral septal nucleus, ventral part, LV lateral ventricle, STMA bed nucleus of the stria terminalis, medial division, anterior part, VMHSh ventmedial nucleus hypothalamus shell. Source data are available online for this figure.

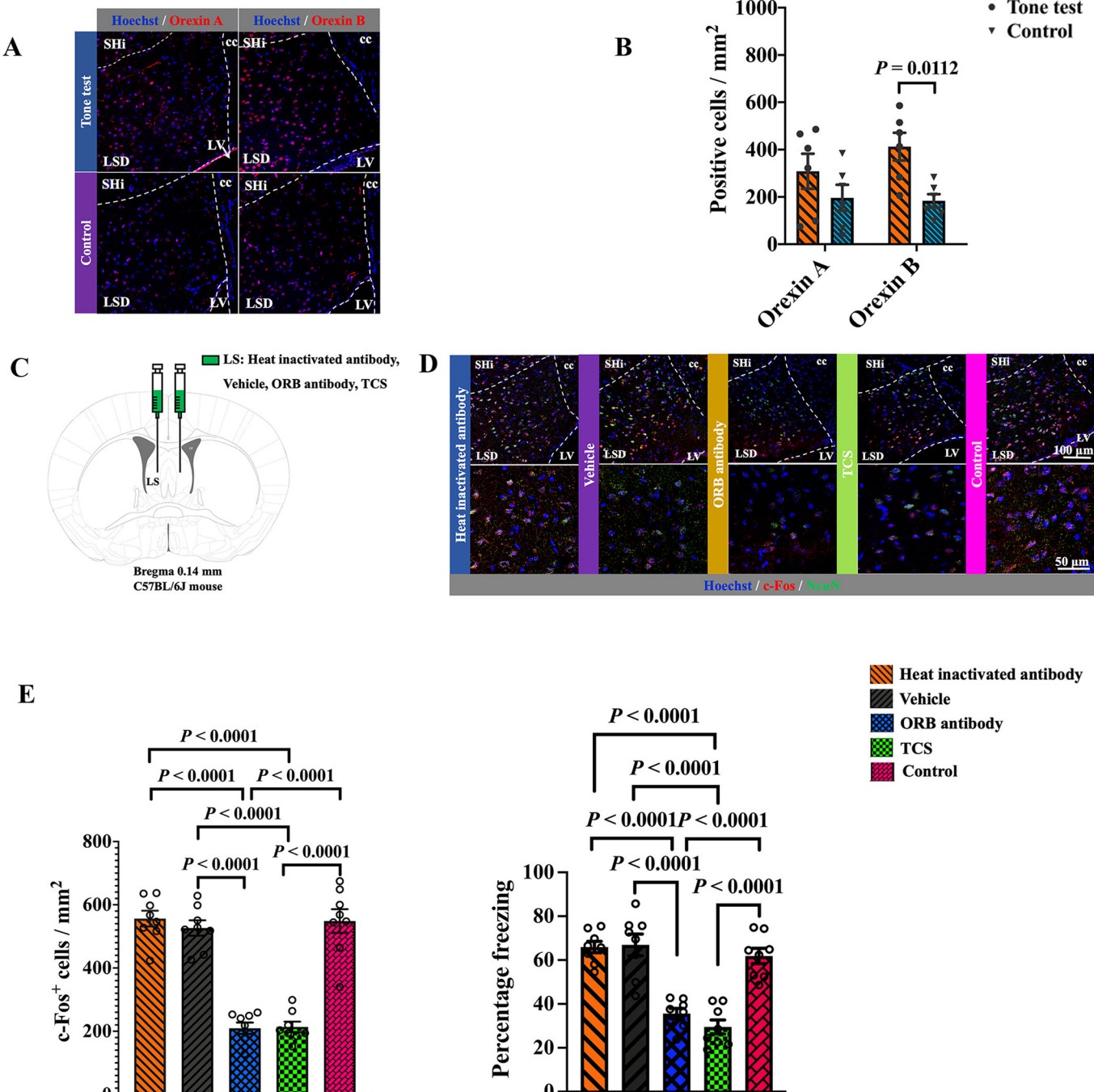

**Figure 8.   Orexin B contributed to tone-related fear conditioning.**

(A) Representative immunohistochemical staining of LS cells, which were labeled by Hoechst (blue) and antibodies recognizing orexin A and orexin B (red), respectively. The dashed lines indicate the boundary of the LS. SHi septohippocampal nucleus, cc corpus callosum, LV lateral ventricle. (B) Number of cells that were positive for orexin A or orexin B in the LS ($n = 6$ mice in each group; $t$ test). Data are presented as mean ± SEM with the presentation of individual animal data in the bar graph. (C) Schematic of the injection of heat-inactivated antibody, vehicle, anti-orexin B antibody or TCS OX2 29 into the LS. Mice were sacrificed for the detection of c-Fos positive cells in the LS at 1 h after memory retrieval tone stimuli. ORB orexin B antibody. (D) Representatives of immunofluorescence staining showing co-localization of c-Fos with NeuN in the LS. Lower panels were high magnification images of the upper panels. The dashed lines indicate the boundary of the LS. SHi septohippocampal nucleus, cc corpus callosum, LV lateral ventricle, NeuN Neuronal Nuclear protein. (E) Left panel: quantification of c-Fos expression in the LS [$n = 8$ mice in each group; $F_{(4,35)} = 51.23$, $P < 0.0001$; Brown–Forsythe and Welch one-way analysis of variance test]. Right panel: performance during fear conditioning [$n = 8$ mice in each group; $F_{(4,35)} = 27.08$, $P < 0.0001$; Brown–Forsythe and Welch one-way analysis of variance test]. Data are presented as mean ± SEM with the presentation of individual animal data in the bar graph. Source data are available online for this figure.

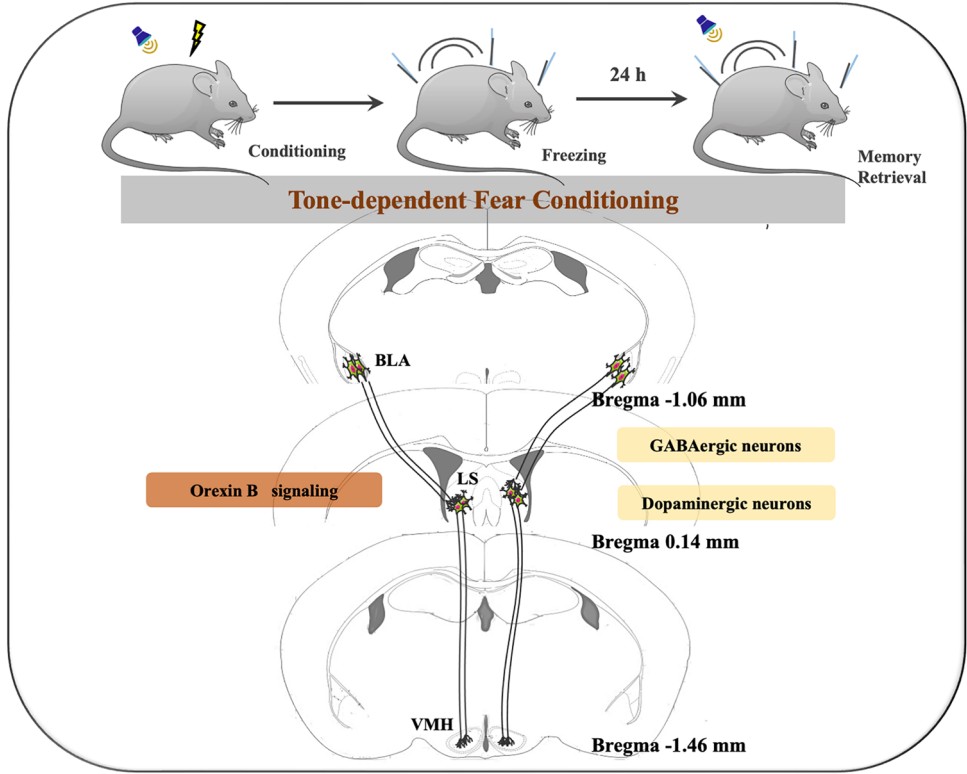

**Figure 9.** Diagram presentation of the role of BLA–LS and LS-VMH neural circuitries in tone-related fear conditioning.

regions to inhibit BLA neurons after the connection between the LS and VMH is suppressed. In this context, it is interesting to note that inhibiting the connection between LS and VMH reduced the number of activated neurons in the VMH. GABAergic neurons are the major neurons in the LS (Risold et al, 1997; Wong et al, 2016). This type of neurons is often considered inhibitory neurons (Xu et al, 2018). Inhibiting inhibitory neurons shall render the activation of downstream neurons. However, GABAergic neurons can be excitatory neurons (Ye et al, 2017). In addition, LS has a large number of dopaminergic neurons that can be excitatory neurons (Chuhma et al, 2004). Also, our study has shown that some neurons in the LS may use both GABA and dopamine as their neurotransmitters. Finally, a possibility is that there are inhibitory interneurons in the VMH. Reducing the activity of inhibitory neurons in the LS activates those inhibitory interneurons, which then inhibits the activation of other neurons in the VMH. These possible mechanisms may explain why inhibiting LS led to inhibition of VMH, a finding in our study.

The context-related fear memory retrieval stimuli and TMT did not increase the number of c-Fos-positive cells in the BLA and VMH in our study, suggesting that these stimuli may not activate these brain regions. BLA is considered an important component for the formation of context-related fear conditioning (Kim et al, 2020; Radulovic et al, 1998) and is activated after the training stimuli (Radulovic et al, 1998). C-Fos increase in the BLA after context-related fear memory retrieval stimuli has been reported (Hammack et al, 2023; Liu et al, 2022) but it may be mostly in the middle of anterior BLA in rodents (Hammack et al, 2023). In addition, a previous study shows no increase in the

number of c-Fos positive cells in the BLA of rats after context-related fear memory retrieval stimuli (Luyck et al, 2020), a result that is similar to our finding. Similar to our findings, TMT induces freezing behavior in rodents (Janitzky et al, 2015). TMT increases the number of c-Fos mRNA-positive cells in multiple brain regions including the LS. However, BLA and BMH are not on the list of brain regions that have increased c-Fos mRNA cells after TMT exposure (Janitzky et al, 2015). Thus, additional studies are needed to clearly define the role of BLA in context-related fear memory retrieval. Currently there is no strong indication for a role of BLA and VMH in the TMT-induced fear conditioning.

Our findings may have significant implications. Our results suggest a critical role of the neural circuitries BLA–LS and LS-VMH in tone-related fear conditioning. These findings assign novel neurobiological functions to these brain regions and to this circuitry. Although it remains to be limited in certain diseases, such as Parkinson's disease (Groiss et al, 2009), in clinical practice to selectively inhibit or activate a brain region, it may be possible clinically to inhibit the BLA–LS and LS-VMH circuitries to reduce the fear and anxiety of patients with various diseases, such as post-traumatic stress disorder.

Our study has limitations. First, there are many subtypes of GABAergic neurons in the LS (Zhao et al, 2013). Determining which subtypes of GABAergic neurons are involved in tone-related fear conditioning will be performed in the future study. We used compound 21 in the chemogenetic experiments. Compound 21 is more effective in activating human Gi-coupled M4 muscarinic receptor and has better penetration to the brain than clozapine-N-

oxide (CNO), another often used agent to achieve chemogenetic regulation (Jendryka et al, 2019). Both CNO and compound 21 can bind competitively various receptors, such as G protein-coupled receptors. However, compound 21 at 3 mg/kg used in this study was not found to affect mouse behavior (Jendryka et al, 2019) but compound 21 at 1 mg/kg causes diuresis in mice in another study (MacIver et al, 2024). In our study, compound 21 in mice receiving control viruses to the LS did not affect the tone-related fear conditioning (Figs. 2 and 6). Thus, the effects observed with inhibiting the LS neurons or the BLA–LS or LS-VMH connections on tone-related fear conditioning are not due to the off-target effects of compound 21.

In summary, we have shown that tone stimuli in the fear conditioning test activate the neural circuitries BLA–LS and LS-VMH, which mediates tone-related fear conditioning. The GABAergic and dopaminergic neurons in the LS and the orexin signaling in the LS neurons are critical for the development of tone-related fear conditioning and fear memory retrieval (Fig. 9).

# Methods

### Reagents and tools table

| Reagent or resource | Source | Identifier |
| --- | --- | --- |
| **Antibodies** | | |
| Rabbit c-Fos polyclonal primary antibody | Abcam | Catalog # ab190289 |
| Goat c-Fos polyclonal primary antibody | Abcam | Catalog # ab 156802 |
| Mouse c-Fos polyclonal primary antibody | Invitrogen | Catalog # MA5-17076 |
| Mouse Anti-VGluT1 antibody | Abcam | Catalog # ab 242204 |
| Rabbit Anti-GAD65 + GAD67 antibody | Abcam | Catalog # ab 11070 |
| Rabbit Anti-GABA antibody | MilliporeSigma | Catalog # A2052 |
| Rabbit Anti-Tyrosine Hydroxylase antibody | Abcam | Catalog # ab 112 |
| Mouse Anti-Tyrosine Hydroxylase antibody | Cell signaling | Catalog # 45648 |
| Mouse Anti-MAP2 antibody | Abcam | Catalog # ab 11267 |
| Rabbit Anti-MAP2 antibody | Abcam | Catalog # ab 32454 |
| Normal Mouse IgG | Santa Cruz | Catalog # sc-2025 |
| Normal Rabbit IgG | Cell Signaling | Catalog # 2729 |
| Rabbit anti-Mouse IgG (H + L) Cross-Adsorbed Secondary Antibody, Alexa Fluor 488 | Invitrogen | Catalog # A-11059 |
| Rabbit anti-Mouse IgG (H + L) Cross-Adsorbed Secondary Antibody, Alexa Fluor 647 | Invitrogen | Catalog # A-21239 |
| Goat anti-Mouse IgG (H + L) Highly Cross-Adsorbed Secondary Antibody, Alexa Fluor Plus 488 | Invitrogen | Catalog # A32723 |
| Goat anti-Mouse IgG (H + L) Highly Cross-Adsorbed Secondary Antibody, Alexa Fluor Plus 594 | Invitrogen | Catalog # A32742 |

| Reagent or resource | Source | Identifier |
| --- | --- | --- |
| Donkey anti-mouse IgG (H + L), Alexa Fluor 647 | Invitrogen | Catalog # A31571 |
| Donkey anti-rabbit IgG (H + L), Alexa Fluor 488 | Invitrogen | Catalog # A21206 |
| Donkey anti-mouse IgG (H + L), Alexa Fluor 488 | Invitrogen | Catalog # A20202 |
| Donkey anti-rabbit IgG (H + L), Alexa Fluor 647 | Invitrogen | Catalog # A31573 |
| Rabbit Anti-NeuN antibody | Abcam | Catalog # ab128886 |
| Goat Anti-Rabbit IgG H&L (HRP) | Abcam | Catalog # ab205718 |
| Anti-Orexin A antibody | Abcam | Catalog # ab6214 |
| Anti-Orexin B antibody | Abcam | Catalog # ab89888 |
| Anti-Cre recombinase antibody | Abcam | Catalog # ab216262 |
| Anti-Cre recombinase antibody | Cell Signaling | Catalog #15036s |
| **Bacterial and virus strains** | | |
| pAAV-hSyn-hM4D(Gi)-mCherry | Addgene | Catalog # 50475-AAV5 |
| pAAV-hSyn- mCherry | Addgene | Catalog # 114472-AAV5 |
| pAAV-CAG-GFP | Addgene | Catalog # 37825-AAVrg |
| pAAV-CAG-tdTomato | Addgene | Catalog # 59462-AAV1 |
| CAV hSyn DIO-hM4D-mCherry (CAV2-Cre) | PVM | N/A |
| pAAV-DIO-hSyn-hM4D(Gi)-mCherry | Addgene | Catalog # 44362-AAV5 |
| pAAV-hSyn-DIO-mCherry | Addgene | Catalog # 50459-AAV5 |
| AAV.rTH.PI.Cre.SV40 | Addgene | Catalog #107788-AAV9 |
| AAV9-mGAD2-Cre | Genecopoeia | Catalog # AA09-CS-CRE-01-200 |
| **Chemicals and drugs** | | |
| Normal Donkey Serum | Sigma -Aldrich | Catalog # D9663 |
| DREADD agonist 21 (Compound 21) | hello bio | Catalog # HB4888 |
| Hoechst | Sigma -Aldrich | Catalog # 23491-45-4 |
| Pierce™ Dimethylsulfoxide (DMSO) | Thermo Fisher SCIENTIFIC | Catalog # 20688 |
| Ketamine | Zoetis | Catalog # KET-00002R2 |
| Xyeline | Fisherbrand | Catalog # HC7001GAL |
| Sucrose | Fisher Chemical | Catalog # 187063 |

| Reagent or resource | Source | Identifier |
|---|---|---|
| Tween 20 | BIO-RAD | Catalog # 1706531 |
| Tris HCl | Fisher Chemical | Catalog # 195295 |
| Sodium Citrate Dihydrate | Fisher Chemical | Catalog # 188858 |
| Triton X-100 | SIGMA | Catalog # SLBV4122 |
| Albumin from bovine serum (BSA) | Sigma -Aldrich | Catalog # SLBD6158V |
| α-Methyl-L-tyrosine | Sigma-Aldrich | Catalog # 672-87-7 |
| DL-2-Allylglycine | Sigma-Aldrich | Catalog # 7685-44-1 |
| UltraPure™ Distilled Water | Invitrogen | Catalog # 2052086 |
| Albumin Standard | Thermo Scientific | Lot # SA242714A |
| TCS OR2 29 | Abcam | Catalog # ab 141316 |
| Immu-Mount | Thermo Scientific | Catalog # 9990402 |
| Ethanol 200 Proof | Decon Labs, Inc | Catalog # 64-17-5 |
| DPBS (1X) | Invitrogen | Catalog # 1929945 |
| PBS (10X) | Life Technologies | Catalog # 2185889 |
| Tissue-Tek O.C.T. Compound | SAKURA | Catalog # 4583 |
| 2,5-dihydro-2,4,5-trimethylthiazoline (TMT) | BioSRQ LLC | Catalog # SRQ-023 |
| **Experimental models** | | |
| C57BL/6J mice | Charles River | N/A |
| **Software and algorithms** | | |
| Fear conditioning software | TSE systems | N/A |
| Stereotaxic Frame | ASI Instruments | N/A |
| Neuros Syringes | Hamilton | Part #/Ref: 65458-01 |
| Calibrated Syringe | Hamilton | Part #/Ref: CAL80135 |
| Graphpad prism 8.0 | http://www.Graphpad.com | N/A |
| Image J | https://imagej.en.softonic.com/mac | N/A |
| Photoshop | | N/A |

## Animals

All procedures were approved by the Institutional Animal Care and Use Committees of the University of Virginia (protocol number: 3114, to Zhiyi Zuo's laboratory) and conducted in accordance to the US National Institute of Health (NIH) guidelines for the handling and use of laboratory animals. Male C57BL/6J mice (Charles River) weighing 18–25 g and at 6–8 weeks of age at the start of experiments were group-housed (3–5 mice per cage) under standard laboratory conditions (12/12 h light/dark cycle, lights on

at 06:00 a.m., $24 \pm 1\,°C$ with a relative humidity of 30% to 70%, food and water available ad libitum) in transparent polycarbonate cages ($16 \times 22 \times 14$ cm) until described otherwise. All experimental procedures were performed between 08:00 a.m. and 03:00 p.m. No statistical methods were used to predetermine the sample size for each experiment. However, sample sizes were consistent with those generally employed in the field. Animals were randomly assigned into experimental groups for each set of experiments and tested in a blind fashion for the experimental conditions. Mice in the control group for the results presented in Fig. 1 were not exposed to preconditioning stimuli or tone-related memory retrieval stimuli (naive mice). The freezing behavior of mice during fear conditioning tests was scored by an individual who was blind to the group assignment of the animals. This paper was written to be in compliance with the ARRIVE guidelines.

## Viral vectors

AAV-hSyn-hM4D(Gi)-mCherry (100 μL at a titer $\geq 7 \times 10^{12}$ vg/mL), AAV-hSyn- mCherry (100 μL at titer $\geq 7 \times 10^{12}$ vg/mL), AAV-CAG-GFP (100 μL at titer $\geq 7 \times 10^{12}$ vg/mL), AAV-CAG-tdTomato (100 μL at a titer $\geq 5 \times 10^{12}$ vg/mL), AAV-DIO-hSyn-hM4D(Gi)-mCherry (100 μL at a titer $\geq 7 \times 10^{12}$ vg/mL), AAV-hSyn-DIO-mCherry (100 μL at a titer $\geq 7 \times 10^{12}$ vg/mL) and AAV.rTH.PI.Cre.SV40 (100 μL at a titer $\geq 1 \times 10^{13}$ vg/mL) were produced by the Addgene Company (Watertown, MA) and used as in our previous studies (Xin et al, 2022; Zeng et al, 2021). CAV2-Cre (CAV hSyn DIO-hM4D-mCherry, 50 μL at a titer $2.5 \times 10^{12}$ vg/mL) was purchased from Plateforme de Vectorologie de Montpellier (France). AAV9-mGAD2-Cre (100 μL at a titer $\geq 1.82 \times 10^{13}$ vg/mL) was custom AAV particles for Cre with the mouse Gad2 promoter and was prepared by Genecopoeia Inc. (Rockville, MD). All viral vectors were stored in aliquots at $-80\,°C$ until use.

## Fear conditioning paradigm

All fear conditioning tests were performed between 8:00 am to 10:00 am. Mice were initially handled and habituated to the conditioning cage ($18\,cm \times 18\,cm \times 30\,cm$) that had an electrifiable floor connected to a H13-15 shock generator (Coulbourn Instruments, Whitehall, PA). This test cage was located inside a sound-attenuated cabinet (H10-24A; Coulbourn Instruments). Before each test session, the test cage was wiped clean with 70% ethanol. During the test, the cabinet was illuminated and the behavior was captured with a monochrome CCD-camera (Panasonic WV-BP334) at 3.7 Hz and stored on a personal computer. The FreezeFrame software (Coulbourn Instruments) was used to control the delivery of both tones and foot shocks. For habituation, three 4-kHz 75-dB tones, each of which was 30 s in duration with a 60 s interval, were delivered. During conditioning, mice received three pairs of the tone and 2-s 0.7-mA foot shock. The test for fear memory was performed 24 h after the conditioning process. The context-related memory was evaluated by placing the mice for 6 min in the same cage where they were exposed to conditioning stimuli. The tone-related memory was tested in a novel illuminated context, where mice were exposed to three tone stimuli. The novel context was a cage ($22\,cm \times 22\,cm \times 21\,cm$) with a different shape and floor texture compared with the conditioning cage. Prior to each use, the floor and walls of the cage were wiped clean with 0.5% citric acid to

make the scent distinct from that of the conditioning cage. The animals were placed in this cage for 3 min before the tone stimuli were turned on. Behavioral responses to the tone stimuli in the new cage were recorded. Freezing behavior was analyzed with FreezeFrame (Coulbourn Instruments).

## TMT stimulation experiment

Mice were moved to the testing room at least 30 min before the start of the experiment. The animal cages were placed in a room next to the testing room to avoid the effect of leaking TMT during the experiment. Before the experiment, a folded Kimwipe was added to the bottom of the transparent testing chamber. To observe the baseline level of freezing behavior, mouse was gently placed into the testing chamber to record the "no-odor" interval lasting for 4 min. Then, 5 µl TMT (Hacquemand et al, 2010) was added to the Kimwipe to observe the TMT-induced freezing behavior for 5 min. After the recording, the mouse was removed from the chamber and placed in a separate cage in another room to prevent untested and being testing mice from being disturbed by TMT. Before testing the next mouse, the testing chamber was wiped with 75% alcohol and put into the fume hood to eliminate the residual odor inside the chamber.

## Stereotaxic surgery

Mice received a subcutaneous injection of 100 mg/kg ketamine and 5 mg/kg xylazine and were briefly head-restrained. Viral injections were performed as we and others have described (Xin et al, 2022; Zeng et al, 2021) with the following stereotaxic coordinates: LS: 0.90 mm from Bregma, 0.40 mm lateral from midline, and 3.80 mm vertical from the cortical surface; VMH: −1.50 mm from Bregma, 0.50 mm lateral from midline, and 5.50 mm vertical from the cortical surface; BLA: –1.50 mm from Bregma, 3.00 mm lateral from midline, and 4.80 mm vertical from the cortical surface. Animals were kept on a heating pad throughout the entire surgical procedures and were brought back to their home cages after 6-h post-surgery recovery and monitoring. Postoperative care included intraperitoneal injection with 0.3–0.5 ml of lactated Ringers solution and Metacam (meloxicam, 1–2 mg/kg) for analgesia and anti-inflammatory purposes. All AAVs and the CAV2-Cre were injected at a total volume of ~0.6 µl (0.3 µl for each side), and were allowed four weeks for maximal expression.

## Chemogenetic manipulations and neural tracing

For chemogenetic manipulation of LS, C57BL/6J mice were bilaterally injected with the AAV-hSyn-hM4Di-mCherry or AAV-hSyn-mCherry. Four weeks later, mice were injected intraperitoneally (i.p.) with compound 21 (3 mg/kg) 30 min before either fear conditioning training or fear memory retrieval tone stimuli. For anterograde and retrograde tracing from LS, C57BL/6J mice were unilaterally injected with the AAV-CAG-tdTomato into the right LS and AAV-CAG-GFP into the left LS. To determine the role of the connection between BLA and LS or between LS and VMH, C57BL/6J mice were bilaterally injected with the CAV2-Cre virus into the LS and then with the AAV-DIO-hSyn-hM4Di-mCherry or AAV-DIO-hSyn-mCherry into the BLA or CAV2-Cre virus into the VMH and then with the AAV-DIO-hSyn-hM4Di-

mCherry or AAV-DIO-hSyn-mCherry into the LS. Four weeks later, mice were injected i.p. with compound 21 30 min before either fear conditioning training or fear memory retrieval tone stimuli. To specifically targeting GABAergic or dopaminergic neurons in the LS, C57BL/6J mice were bilaterally injected with AAV-hSyn-DIO-hM4D(Gi)-mCherry plus AAV.rTH.PI.Cre.SV40 or AAV-hSyn-DIO-hM4D(Gi)-mCherry plus AAV9-mGAD2-Cre in the LS. Four weeks later, mice were injected i.p. with compound 21 30 min before fear memory retrieval tone stimuli were applied.

## Intracerebral injection

Mice were anesthetized by ketamine and xylazine as described in section 4.4. DL-2-allylglycine [Sigma-Aldrich, solubilized in phosphate-buffered saline (PBS) to 100 mM and in 10% dimethyl-sulfoxide (DMSO) to 10 mM], or α-methyl-L-tyrosine (Sigma-Aldrich, solubilized in PBS to 100 mM and in 10% DMSO to 10 mM), or PBS + 10% DMSO (Thermo Fisher Scientific) as the control was slowly injected bilaterally into the LS at a flow-rate of 0.5 µl/min and in a total volume of 0.3 µl per injection site. Following the injection, the needle was left in place for 3 min to allow the solution to diffuse away from the needle tip. Injection procedure of the anti-orexin B antibody, heat inactive anti-Orexin B antibody and TCS OX2 29 (ab141316, Abcam, solubilized in water to 100 mM and in DMSO to 10 mM) was the same as described for DL-2-allylglycine. After the surgery, all animals received a subcutaneous injection of 3 mg/kg bupivacaine. Mice were allowed to recover from the injection procedure for a minimum of three days, and then the fear conditioning test was performed.

## Immunohistochemistry

Animals were deeply anesthetized with isoflurane for 2 min and transcardially perfused with PBS, followed by perfusion with 4% paraformaldehyde in PBS. The brains were quickly extracted and incubated in 4% paraformaldehyde overnight. This was followed by cryoprotection in a 30% PBS-buffered sucrose solution until brains were saturated (24–36 h). Coronal brain sections (50 µm for the whole brain sections and 20 µm for other experiments) were cut using a cryostat (SM 2010R, Leica). Brain sections were washed in PBST (0.1% Triton X-100 in PBS, 3 × 15 min) at room temperature (RT). Next, sections were blocked in 10% normal donkey serum (NDS) in PBST for 3 h at RT, followed by incubation with primary antibodies overnight at 4 °C. Sections were then washed with PBST (3 × 15 min) and incubated with fluorescent secondary antibodies at RT for 1 h. After being washed with PBS (3 × 15 min), sections were mounted onto glass slides with Fluoromount (Sigma-Aldricht). Images were taken using a LSM 880 laser-scanning confocal microscope (Carl Zeiss). The primary antibodies used were: anti-c-Fos (1:1000, rabbit, Abcam, catalog number: ab190289); anti-vGluT1 antibody (1:5000, mouse, Abcam, catalog number: ab 242204), anti-GAD65&67 antibody (1:200, rabbit, Abcam, catalog number: ab 11070); anti-tyrosine hydroxylase antibody (1:750, rabbit, Abcam, catalog number: ab 112, or 1:200, mouse, Cell Signaling, catalog number: 45648); anti-GABA (1:500, rabbit, MilliporeSigma, catalog number: A2052), anti-microtubule-associated protein 2 (MAP2) antibody (1:200, mouse, Abcam, catalog number: ab 11267 and 1:200, rabbit, Abcam, catalog

number: ab 32454); anti-neuronal nuclear protein (NeuN) antibody (1:1000, rabbit, Abcam, catalog number: ab128886); anti-orexin A antibody (1:1000, rabbit, Abcam, catalog number: ab6214); anti-orexin B antibody (1:100, mouse, Abcam, catalog number: ab89888); and anti-Cre recombinase antibody (1:1000, rabbit, Abcam, catalog number: ab216262, or 1:200, rabbit, Cell Signaling, catalog number: 15036s). Fluorophore-conjugated secondary antibodies were purchased from Invitrogen. Antibodies were diluted in PBS with 0.1% Triton X-100.

### c-Fos⁺ cell counting in the brain sections

The construction of the heatmap of regions with c-Fos expression in mice with or without tone stimuli was performed as we reported previously (Xin et al, 2022; Zeng et al, 2021). Briefly, frozen coronal brain sections of 50-μm thickness were cut sequentially from the whole brain and one section from ten sections was stained for c-Fos. The number of c-Fos positive cells in three randomly selected and independent microscopic fields in each brain structure of interest in each section and three sections for each mouse was counted. The average of the nine values from each mouse was used to reflect the level of c-Fos expression in the brain region of the mouse and presented as the number of c-Fos-positive cell/mm². 

To determine c-Fos expression in the LS, BLA and VMH after various experimental conditions, we randomly selected 3 non-consecutive 20-μm thick coronal sections between 0.62 and 1.10 mm from the Bregma along the rostral-caudal axis for the LS, −1.22 and −1.70 mm from the Bregma along the rostral-caudal axis for the BLA and VMH from each mouse and stained them for c-Fos. High-resolution 15-μm stacks of these structures were acquired and projected along the z-axis using a LSM 880 confocal microscope (Zeiss). We identified the LS based on forebrain hallmarks (lateral ventricle), or BLA and VMH based on hypothalamic hallmarks, and manually counted c-Fos-positive cells in these regions. These cells needed to be positive for Hoechst staining for them to be counted. Results were averaged across bilateral regions and sections for each mouse for preparing the heatmap of c-Fos expression.

### Statistics and data presentation

No mice or data points were excluded from analysis. All data were presented as mean ± SEM (normal distribution data) or median ± interquartile range (not normal distribution data) with the presentation of data of each individual animal in the bar graphs. Normality of the data was examined by Shapiro–Wilk test. The results were analyzed by using Student's *t* test, one-way analysis of variance followed with Tukey test, rank-sum test, one-way analysis of variance on ranks followed with Tukey test as appropriate. The statistical method used for the dataset presented in each figure was described in the figure legend. Significant differences were accepted at a $P < 0.05$ based on two-tailed hypothesis testing. The exact $P$ value for the comparison between the two groups was presented in the figures.

## Data availability

This study includes no data deposited in external repositories.

The source data of this paper are collected in the following database record: biostudies:S-SCDT-10_1038-S44319-025-00403-x.

## Peer review information

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

## Acknowledgements

This study was supported by the Robert M. Epstein Professorship endowment (to Z Zuo) and the Department of Anesthesiology, University of Virginia, Charlottesville, VA, USA. All studies described in the manuscript were performed in the University of Virginia. We would like to thank Dr. Chia-Yi (Alex) Kuan, Department of Neuroscience, University of Virginia, for constructive discussion and for allowing us to use his cryostat.

## Author contributions

**Miao Chen**: Data curation; Formal analysis; Validation; Investigation; Visualization; Methodology; Writing—original draft. **Jun Li**: Data curation; Formal analysis; Validation; Investigation; Visualization; Methodology. **Weiran Shan**: Data curation; Formal analysis; Validation; Investigation; Visualization; Methodology; Writing—original draft. **Jianjun Yang**: Investigation. **Zhiyi Zuo**: Conceptualization; Resources; Formal analysis; Supervision; Funding acquisition; Validation; Visualization; Writing—original draft; Project administration; Writing—review and editing.

Source data underlying figure panels in this paper may have individual authorship assigned. Where available, figure panel/source data authorship is listed in the following database record: biostudies:S-SCDT-10_1038-S44319-025-00403-x.

## Disclosure and competing interests statement

The authors declare no competing interests.

