## [Peer Review File · EMBO Reports]

Auditory fear memory retrieval requires BLA-LS and LS-VMH circuitries via GABAergic and dopaminergic neurons

Miao Chen, Jun LI, Weiran Shan, Jian-Jun Yang, and Zhiyi Zuo

Corresponding author(s): Zhiyi Zuo (zz3c@virginia.edu)

Review Timeline:

Submission Date:	28th Feb 24
Editorial Decision:	18th Apr 24
Revision Received:	24th Dec 24
Editorial Decision:	21st Jan 25
Revision Received:	6th Feb 25
Accepted:	12th Feb 25

Editor: Esther Schnapp

Transaction Report:

Dear Dr. Zuo,

Thank you for the submission of your manuscript to EMBO reports. We have now received the full set of referee reports as well as referee cross-comments that are all pasted below.

As you will see, the referees acknowledge that the findings are potentially interesting. However, both referees 1 and 2 point out that additional and stronger data will be required to make this study a good fit for EMBO reports. Referee 1 notes that further insight into the roles that the pathways in the LS may play in the retention of fear memory would be required, and referee 2 adds that it is nowadays common practise to perform optogenetic/chemogenetic manipulations to achieve a more selective/accurate control of neuronal activity compared with pharmacological interventions.

Given that you informed us that you can address all referee comments, and that referee 1 is satisfied with the way you plan to address them, I am happy to invite you to revise your manuscript with the understanding that the referee concerns must be fully addressed and their suggestions taken on board. Please address all referee concerns in a complete point-by-point response.

Acceptance of the manuscript will depend on a positive outcome of a second round of review. It is EMBO reports policy to allow a single round of major revision only and acceptance or rejection of the manuscript will therefore depend on the completeness of your responses included in the next, final version of the manuscript.

We realize that it is difficult to revise to a specific deadline. In the interest of protecting the conceptual advance provided by the work, we recommend a revision within 3 months (19th Jul 2024). Please discuss the revision progress ahead of this time with the editor if you require more time to complete the revisions.

- 1) A data availability section providing access to data deposited in public databases is missing. If you have not deposited any data, please add a sentence to the data availability section that explains that.
- 2) Your manuscript contains statistics and error bars based on $n=2$. Please use scatter blots in these cases. No statistics should be calculated if $n=2$.

3) We replaced Supplementary Information with Expanded View (EV) Figures and Tables that are collapsible/expandable online. A maximum of 5 EV Figures can be typeset. EV Figures should be cited as 'Figure EV1, Figure EV2' etc... in the text and their respective legends should be included in the main text after the legends of regular figures.

5) a complete author checklist, which you can download from our author guidelines <https://www.embopress.org/page/journal/14693178/authorguide>. Please insert information in the checklist that is also reflected in the manuscript. The completed author checklist will also be part of the RPF.

6) Please note that all corresponding authors are required to supply an ORCID ID for their name upon submission of a revised manuscript (<https://orcid.org/>). Please find instructions on how to link your ORCID ID to your account in our manuscript

tracking system in our Author guidelines

<<https://www.embopress.org/page/journal/14693178/authorguide#authorshippinguidelines>>

10) Regarding data quantification (see Figure Legends:

<https://www.embopress.org/page/journal/14693178/authorguide#figureformat>)

- the name of the statistical test used to generate error bars and P values,
- the number (n) of independent experiments (please specify technical or biological replicates) underlying each data point,
- the nature of the bars and error bars (s.d., s.e.m.),
- If the data are obtained from $n < 3$, please use scatter blots showing the individual data points.

I look forward to seeing a revised form of your manuscript when it is ready.

Referee #1:

The manuscript by Chen et al. presents a series of experiments aimed at investigating the involvement of the lateral septum (LS) in auditory fear learning and memory retrieval. The experiments are properly conducted and analyzed. However, I have three main concerns regarding the obtained results.

Firstly, it is not clear from the methods and legends which controls were performed in the *cfos* expression analysis. To support the claim that the increase in *cfos* observed in the tone fear conditioning group is related to tone fear memory retrieval, it would be necessary to compare this data with that obtained in groups of subjects i) untrained (naive), ii) exposed to sound alone, and iii) subjected to shock-only stimulation.

Secondly, contrary to what the authors assert in the abstract and introduction, previous studies have already shown the involvement of LS in tone fear learning and memory. Some studies have indeed shown that LS activity increases after tone fear learning and memory retention (Butler CW et al. *Learn. Mem.* 2015; Holschneider DP et al., *Neuroimage* 2006; Garcia and Jaffard, *Eur. J. Neurosci.* 1996), while others have also demonstrated that this site is necessary for tone fear learning (among others, the studies by A. Desmedt and colleagues in the 2007 and 2010). These studies should be presented from the outset, removing incorrect statements such as "However, its role in tone-related fear conditioning has not been reported yet." Regarding the aforementioned studies, the main novelty of the present work is to have identified two necessary pathways for tone fear memory: namely, one from the basolateral amygdala (BLA) terminating in LS and the other from LS terminating in the ventromedial hypothalamus (VMH). However, it is not clear what role these pathways may play in fear memory retention: are they necessary to maintain the CS-US association? To produce freezing responses and/or aversion in general? Does blocking these pathways also compromise the retention of contextual fear memory information? Indeed, no experiments were conducted by the authors to try to answer these questions.

The last point I have is related to the concept of "pathway." The authors have indeed identified a pathway from BLA to LS and one from LS to VMH. However, whether these two pathways are anatomically and functionally connected has not been investigated at all. Therefore, it is not appropriate to identify these two pathways as BLA-LS-VMH as the authors do throughout the manuscript.

Therefore, considering the points just described, it is not clear if and in what terms the data presented in this study can represent a major advance in the field.

Referee #2:

This study was designed to determine the role of LS in the development of tone-related fear conditioning. The authors reported that the basolateral amygdala (BLA)→LS→ventromedial nucleus of the hypothalamus (VMH) neural circuitry is critical for the tone-related fear conditioning. Furthermore, GABAergic and dopaminergic neurons that are activated by orexin B in the LS play a role in this fear conditioning. The results are potentially interesting; however, several concerns remain to be addressed.

(1) Fig. S2A. The authors showed a large number of TH+ neurons in the LS, and claimed that there are no glutamatergic neurons in this brain region. However, Li et al. (2023) (PMID: 35994589) showed the prominent distribution of CaMKII α + cells in the ventral of lateral septal (LSV) region. Accordingly, there are indeed some glutamatergic neurons in the LS, depending on the specific subregion analyzed. In addition, it is kind of surprising to see so many TH+ dopaminergic cells in the LS. What is the relative percentage of the TH+ cells among all LS cells? Are there any TH+/GAD65+ co-labeling neurons?

(2) Fig. 1D. Please provide the *c-fos* data across different bregma planes of the LS. This will demonstrate the specific LS subregion potentially involved in tone-related fear conditioning.

(3) Fig. 3C-E. To determine the role of LS GABAergic neurons and dopaminergic neurons in tone-evoked fear, it is better to use the chemogenetic or optogenetic approaches to achieve selective inhibition of these divergent cell types in the LS. Although infusion of GAD inhibitor or TH inhibitor will block the synthesis of GABA and dopamine, respectively, such manipulation could not directly demonstrate the role of GABAergic neurons and dopaminergic neurons.

(4) LS consists of multiple subtypes of GABAergic neurons with different molecular markers, such as *sst*, *crhr2*, *nts*, etc. Then which subtype of GABAergic neurons contributes to the tone-evoked fear?

(5) Fig. 5D and F. Since LS sends inhibitory projections to the VMH, it is unreasonable to see a decreased c-fos expression in the VMH following chemogenetic inhibition of the BLA→LS→VMH circuitry.

(6) Fig. 6. What is the input source of the orexin B? Which brain region innervates the LS to release orexin B?

Referee #3:

This is a comprehensive study showing that tone-related fear conditioning is mediated by the neural circuitry involving basolateral amygdala (BLA), lateral septum (LS), and ventromedial nucleus of the hypothalamus (VMH). The study also identifies the types of neurons (GABAergic neurons and dopaminergic neurons) and the neurotransmitter (orexin signaling) in the LS that are responsible for this effect. Given the limited understanding of tone-related fear conditioning and its importance in the basic neurobiology of defensive behavior development, as well as its implications for various psychological and psychiatric illnesses such as post-traumatic stress disorder, this study is both significant and timely. The study presents sufficient evidence to support the conclusion, and the analyses and results are clearly explained. The heat map depicting the brain regions that were activated by tone stimuli and the connections of LS with other brain regions represent a large amount of work and will be very useful to investigators in related fields. I have only a few minor points for the authors to consider.

- 1) Authors may further discuss why they chose to study BLA-LS and LS-VMH connections. For example, they could elaborate on whether these connections have been previously investigated for their role in tone-related fear conditioning.
- 2) Please specify when fear conditioning tests were conducted during the day-were they performed in the morning or afternoon? Additionally, were animals tested within a similar time window each day?
- 3) In Figure 6E, the group labels from the X-axis can be removed and placed on the right hand side of the panel, similar to the group labels in Figure 5. Please also use the same color and filling patterns for the same group across both figures.
- 4) Authors may consider adding a discussion on limitations of the study. For example, potential off-target effects associated with compound 21.

Cross-comments from referee 1:

This study identifies two neural pathways that are activated and necessary during the retention of auditory fear memory. Although it is interesting to identify pathways involved in fear and memory (as countless studies have been doing lately), these results alone do not seem particularly novel and interesting to me, especially in light of previous literature showing the role of the LS in aversion and fear in general, as well as in contextual fear and tone memory. Therefore, to assess the impact of these results and enhance their interest, I believe that anything that can further explore the mechanism and functional significance of these pathways is appropriate. In this context, I completely agree with the experiments suggested by referee 2. Regarding my proposal, the idea is to understand the role that these pathways may play in the retention of fear memory: do they serve to maintain the CS-US association, or do they play a more general role in producing aversion and/or freezing? To discriminate between these possibilities, I believe it would be necessary to at least test the impact of these pathways on contextual fear memory and freezing induced by innate stimuli, such as fox odor, and compare the results obtained to what is already present in the literature regarding all these phenomena.

Cross-comments from referee 2:

I totally agree with the referee 1's comments. Yes, it is indeed the truth that role of LS in fear has been extensively studied in the previous literature. So to improve the novelty of this manuscript, one must check whether LS plays a selective role in retention of tone fear memory or a general role in all types of fear. This point is of vital importance for us to discriminate the exact functions of LS circuits in different forms of fear. Probably, LS GABAergic neurons may equally contribute to contextual fear memory, but through different subsets of neurons or different input-output circuit connections. Regarding my question 3, I think it is nowadays one necessary step to perform optogenetic/chemogenetic manipulations to achieve a more selective accurate control of neuronal activity than the pharmacological interventions.

Responses to the referees' comments

We want to thank the editor and referees for their effort in improving our paper. We have revised the manuscript based on the comments. The changes made to address the comments are in red in the manuscript for easy recognition. Our point-by-point responses to the referees' comments are as follows.

Referee 1

The manuscript by Chen et al. presents a series of experiments aimed at investigating the involvement of the lateral septum (LS) in auditory fear learning and memory retrieval. The experiments are properly conducted and analyzed. However, I have three main concerns regarding the obtained results.

Point 1: *It is not clear from the methods and legends which controls were performed in the cfos expression analysis. To support the claim that the increase in cfos observed in the tone fear conditioning group is related to tone fear memory retrieval, it would be necessary to compare this data with that obtained in groups of subjects i) untrained (naive), ii) exposed to sound alone, and iii) subjected to shock-only stimulation.*

Response: Brains were harvested 1 h after the completion of the 3 tone stimuli (memory retrieval stimuli) (stated in the first sentence of the first paragraph on page 6) to generate data presented in figure 1. These mice were exposed only to sound 1 h before the brain was harvested. The control mice are naïve mice. We used this experimental design because our interest in this study was tone (sound)-related fear conditioning behavior. Thus, we do not feel that it is necessary to compare our data with mice that were exposed to sound alone (this is our intervention group) or with mice that were subjected to shock-only stimulation. Also, not all brain regions with increased c-Fos after the tone stimulation are involved in tone-related fear conditioning. This is why we used chemogenetic approach to determine the involvement of BLA-LS and LS-VMH connections in the tone-related fear conditioning. In the revised manuscript, we have clearly stated that the control group is naïve mice in the methods (first paragraph, page 22) and figure legend of figure 1.

Point 2: *Contrary to what the authors assert in the abstract and introduction, previous studies have already shown the involvement of LS in tone fear learning and memory. Some studies have indeed shown that LS activity increases after tone fear learning and memory retention (Butler CW et al Learn. Mem 2015; Holschneider DP et al., Neuroimage 2006; Garcia and Jaffard, Eur. J. Neurosci. 1996), while others have also demonstrated that this site is necessary for tone fear learning (among others, the studies by A. Desmedt and colleagues in the 2007 and 2010). These studies should be presented from the outset, removing incorrect statements such as "However, its role in tone-related fear conditioning has not been reported yet."*

Response: We agree that previous research has shown the activation of LS neurons after fear learning. However, activation of neurons in a brain region after fear learning may not be sufficient evidence to suggest that these activated neurons in the brain region are involved in the fear conditioning behaviors. We have found 3 reports indicating the role of LS in tone-related fear conditioning based on our search. Two papers on relevant subjects were published by Dr.

Joachim Spiess's group. The first one was published in 1999 (PMID: 10366634) and showed that injecting corticotropin-releasing factor (CRF) into LS before or immediately after the conditioning stimuli reduced tone-related fear conditioning. The second paper in 2007 (PMID: 17553007) showed that immobilization for 1 h (as a stress stimulus) reduced tone-related fear conditioning if the fear conditioning test was performed immediately after the stress stimulus. This effect disappeared if fear conditioning test was started 0.5 h after the stress stimulus. Injecting various CRF agonists and antagonists into LS before the stress stimulus did not affect the attenuated tone-related fear conditioning. Thus, this paper did not provide evidence for the involvement of LS in tone-related fear conditioning. Both papers from Dr. Spiess's group did not document whether neurons in the LS were activated or inhibited after the various manipulations. Two relevant papers on the role of LS in tone-related fear conditioning are from Dr. Aline Demedt's group. The one published in 2007 (PMID: 17554087) showed that infusing lidocaine to LS during application of conditioning stimuli attenuated tone-related fear conditioning. This finding was stated in the Introduction of our manuscript. The second paper published in 2010 (PMID: 20798266) showed that infusion of glutamate or kynurenate (a competitive glutamate antagonist) into LS immediately before the application of conditioning stimuli enhanced (glutamate infusion) or attenuated (kynurenate infusion) tone-related fear conditioning. All three papers (two from Dr. Demedt's group and one from Dr. Spiess's group) do not address whether LS is involved in the tone-related fear memory retrieval, which is a focus of our study. Our study has shown that LS is involved in fear learning and memory retrieval. Thus, our study provides novel finding on the role of LS in fear conditioning.

In the revision of our manuscript, we have cited relevant references and stated their findings based on this literature analysis in paragraph 3, page 4. We have changed the sentence "However, its role in tone-related fear conditioning has not been reported yet." to "However, its role in tone-related fear memory retrieval has not been reported yet."

***Point 3:** Regarding the aforementioned studies, the main novelty of the present work is to have identified two necessary pathways for tone fear memory: namely, one from the basolateral amygdala (BLA) terminating in LS and the other from LS terminating in the ventromedial hypothalamus (VMH). However, it is not clear what role these pathways may play in fear memory retention: are they necessary to maintain the CS-US association? To produce freezing responses and/or aversion in general? Does blocking these pathways also compromise the retention of contextual fear memory information? Indeed, no experiments were conducted by the authors to try to answer these questions.*

Response: In addition to the novelty stated here by the reviewer, we feel that another novel finding from our study is the demonstration of the involvement of LS in fear memory retrieval as stated above. Our results have shown that the connections between BLA to LS or LS to VMH are needed for maintaining the conditioned stimulus-unconditioned stimulus (CS-US) association because disrupting these connections only before the conditioning stimuli impaired tone-related fear conditioning (Figure 6). Our new data of freezing behavior immediately before the memory retrieval stimuli were applied to the mice that produced data in figures 6F and 6H showed that there was no difference in the amount of freezing behavior among the various groups (Fig. 6D), indicating that interrupting these connections does not affect freezing behavior in general. Finally, conditioned contextual fear stimuli induced freezing behavior and activated LS but did

not activate BLA and VMH (Fig. 7), suggesting that the activation of the connections between BLA and LS or LS and VMH is specific for tone-related fear conditioning.

Point 4: *The last point I have is related to the concept of "pathway." The authors have indeed identified a pathway from BLA to LS and one from LS to VHM. However, whether these two pathways are anatomically and functionally connected has not been investigated at all. Therefore, it is not appropriate to identify these two pathways as BLA-LS-VHM as the authors do throughout the manuscript.*

Response: We feel we have evidence to suggest the BLA-LS-VHM pathway based on the anterograde and retrograde tracing results (Figure 5, anatomical connection evidence) and the finding that inhibiting neurons in the BLA reduced the number of activated neurons in the LS and VMH (Figure 6, activity connection evidence). However, we can remove the phrase of BLA-LS-VHM pathway and just use BLA-LS and LS-VHM connections if the referee or editor prefers us to do that.

Point 5: *Therefore, considering the points just described, it is not clear if and in what terms the data presented in this study can represent a major advance in the field.*

Response: In addition to providing evidence for the role of LS in the tone-related fear memory retrieval and the BLA-LS and LS-VHM connections in the tone-related fear conditioning, we have identified molecular mechanisms and types of neurons (the involvement of orexin B signaling, GABAergic neurons and dopaminergic neurons) for tone-related fear conditioning. Finally, our study has shown the brain regions that are activated by fear memory retrieval stimuli and brain regions that are connected with LS. Although some brain regions have been reported, many brain regions have not been described. The work to identify these brain regions represents a large amount of work and is very useful and significant for people who are interested in the neurobiology of fear conditioning and LS.

Referee 2:

This study was designed to determine the role of LS in the development of tone-related fear conditioning. The authors reported that the basolateral amygdala (BLA)→LS→ventromedial nucleus of the hypothalamus (VMH) neural circuitry is critical for the tone-related fear conditioning. Furthermore, GABAergic and dopaminergic neurons that are activated by orexin B in the LS play a role in this fear conditioning. The results are potentially interesting; however, several concerns remain to be addressed.

Point 1: *Fig. S2A. The authors showed a large number of TH+ neurons in the LS, and claimed that there are no glutamatergic neurons in this brain region. However, Li et al. (2023) (PMID: 35994589) showed the prominent distribution of CaMKIIα+ cells in the ventral of lateral septal (LSV) region. Accordingly, there are indeed some glutamatergic neurons in the LS, depending on the specific subregion analyzed. In addition, it is kind of surprising to see so many TH+ dopaminergic cells in the LS. What is the relative percentage of the TH+ cells among all LS cells? Are there any TH+/GAD65+ co-labeling neurons?*

Response: We agree that there is evidence for the existence of glutamatergic neurons in the LS based on the CaMKII α staining. Our failure to detect glutamatergic neurons based on the lack of expression of vesicular GluT1 in two sets of staining experiments in LS may be due to different biomarkers were chosen for glutamatergic neurons or different anatomic locations of brain sections were used between our study and the previous study (PMID: 35994589). Based on this discussion, we have modified our description on whether there are glutamatergic neurons in the LS in the first paragraph, page 7.

Yes, we detected a lot of TH positive cells. Determining the number of TH positive cells accurately may not be easy because TH staining is diffuse in the cytosol and the staining does not provide clear individual cell structure. Also, the number of TH positive cells may vary depending on the locations/regions in the LS. Finally, our results in figures 3 and 4 suggest a role of TH positive cells in the tone-related fear conditioning. Thus, we do not know whether knowing the percentage of TH positive cells will add significantly to what we have shown in the manuscript.

Our co-labeling experiment showed that there were cells that were positive for both GABA and TH staining in the LS (Fig. EV3A), suggesting that these neurons may use more than one neurotransmitter to affect their downstream neurons.

Point 2: Fig. 1D. Please provide the c-fos data across different bregma planes of the LS. This will demonstrate the specific LS subregion potentially involved in tone-related fear conditioning.

Response: As shown in figures 1A and 1B, there was an increase in the c-Fos positive cells in all 3 LS regions, LSD, LSI and LSV, in mice after the fear memory retrieval stimuli compared with naïve mice. Accurate c-Fos data across different bregma planes may be difficult to get because brain tissues on the cutting blocks may be tilted and assigning bregma planes to the sections cut from the blocks may not be accurate. Also, it is difficult to specifically inhibit only a selected region (bregma planes) in the LS by any methods to indicate the role of the selected LS region in tone-related fear conditioning.

Point 3: Fig. 3C-E. To determine the role of LS GABAergic neurons and dopaminergic neurons in tone-evoked fear, it is better to use the chemogenetic or optogenetic approaches to achieve selective inhibition of these divergent cell types in the LS. Although infusion of GAD inhibitor or TH inhibitor will block the synthesis of GABA and dopamine, respectively, such manipulation could not directly demonstrate the role of GABAergic neurons and dopaminergic neurons.

Response: Using GAD inhibitor or TH inhibitor to imply the role of GABAergic neurons and dopaminergic neurons in an effect has been existed in the literature (for example, PMID: 19800953 and PMID: 31870855). To add evidence for the role of GABAergic neurons and dopaminergic neurons in the tone-related fear conditioning, we used a chemogenetic approach where AAV-TH-cre and AAV-DIO-hM4Di-mCherry viruses or AAV-GAD-cre and AAV-DIO-hM4Di-mCherry viruses were injected into LS. Inhibition of GABAergic neurons or dopaminergic neurons in the LS was initiated by compound 21. Inhibiting GABAergic neurons or dopaminergic neurons in the LS before the application of tone-related fear memory stimuli inhibited the increase of activated neurons and freezing behaviors (Fig. 7). These results suggest

that these neurons are important for tone-related fear memory retrieval.

Point 4: *LS consists of multiple subtypes of GABAergic neurons with different molecular markers, such as sst, crhr2, nts, etc. Then which subtype of GABAergic neurons contributes to the tone-evoked fear?*

Response: Our study has provided a large set of data to understand the neural circuitry and molecular mechanisms for tone-related fear conditioning. Studying the role of GABAergic neurons with different biomarkers in the LS in tone-related fear conditioning will be our next project as discussed in the third paragraph, page 14.

Point 5: *Fig. 5D and F. Since LS sends inhibitory projections to the VMH, it is unreasonable to see a decreased c-fos expression in the VMH following chemogenetic inhibition of the BLA→LS→VMH circuitry.*

Response: Although GABAergic neurons are often inhibitory, they can be excitatory (for example, PMID: 29202749). Also, there are many dopaminergic neurons in the LS. Dopaminergic neurons can be excitatory (for example, PMID: 14749442). Thus, it may be difficult to conclude that LS sends only inhibitory projections to the VMH. We have discussed the issue in the first paragraph, page 14.

Point 6: *Fig. 6. What is the input source of the orexin B? Which brain region innervates the LS to release orexin B?*

Response: We did not study the origin of orexin B that was in the LS. Orexin B may be from the lateral hypothalamic area where most orexin neurons are located. It is known that the lateral hypothalamic area sends projects to many brain regions including septum (for example: PMID: 35994589) as discussed in the second paragraph, page 13.

Referee 3:

This is a comprehensive study showing that tone-related fear conditioning is mediated by the neural circuitry involving basolateral amygdala (BLA), lateral septum (LS), and ventromedial nucleus of the hypothalamus (VMH). The study also identifies the types of neurons (GABAergic neurons and dopaminergic neurons) and the neurotransmitter (orexin signaling) in the LS that are responsible for this effect. Given the limited understanding of tone-related fear conditioning and its importance in the basic neurobiology of defensive behavior development, as well as its implications for various psychological and psychiatric illnesses such as post-traumatic stress disorder, this study is both significant and timely. The study presents sufficient evidence to support the conclusion, and the analyses and results are clearly explained. The heat map depicting the brain regions that were activated by tone stimuli and the connections of LS with other brain regions represent a large amount of work and will be very useful to investigators in related fields. I have only a few minor points for the authors to consider.

Point 1: Authors may further discuss why they chose to study BLA-LS and LS-VMH connections. For example, they could elaborate on whether these connections have been previously investigated for their role in tone-related fear conditioning.

Response: We have further discussed why we chose BLA-LS and LS-VMH connections to study in the second paragraph, page 8.

Point 2: Please specify when fear conditioning tests were conducted during the day-were they performed in the morning or afternoon? Additionally, were animals tested within a similar time window each day?

Response: All behavior tests were performed between 8:00 am to 10:00 am. This information is added as the first sentence in the third paragraph, page 22.

Point 3: In Figure 6E, the group labels from the X-axis can be removed and placed on the right hand side of the panel, similar to the group labels in Figure 5. Please also use the same color and filling patterns for the same group across both figures.

Response: We have corrected the figure based on your suggestions.

Point 4: Authors may consider adding a discussion on limitations of the study. For example, potential off-target effects associated with compound 21.

Response: The potential off-target effects of compound 21 are discussed from the last paragraph, page 14, to the first paragraph, page 15.

Cross-comments from referee 1:

This study identifies two neural pathways that are activated and necessary during the retention of auditory fear memory. Although it is interesting to identify pathways involved in fear and memory (as countless studies have been doing lately), these results alone do not seem particularly novel and interesting to me, especially in light of previous literature showing the role of the LS in aversion and fear in general, as well as in contextual fear and tone memory. Therefore, to assess the impact of these results and enhance their interest, I believe that anything that can further explore the mechanism and functional significance of these pathways is appropriate. In this context, I completely agree with the experiments suggested by referee 2. Regarding my proposal, the idea is to understand the role that these pathways may play in the retention of fear memory: do they serve to maintain the CS-US association, or do they play a more general role in producing aversion and/or freezing? To discriminate between these possibilities, I believe it would be necessary to at least test the impact of these pathways on contextual fear memory and freezing induced by innate stimuli, such as fox odor, and compare the results obtained to what is already present in the literature regarding all these phenomena.

Response: Our new data showed that conditioned contextual fear stimuli or 2,5-dihydro-2,4,5-trimethylthiazoline (TMT) induced freezing behavior and activated LS but did not activate BLA

and VMH (Fig. 7), suggesting that the activation of the connections between BLA and LS or LS and VMH is specific for tone-related fear conditioning. Please also see our response to your point 3 for additional information regarding the specific involvement of these connections for tone-related fear conditioning.

Cross-comments from referee 2:

I totally agree with the referee 1's comments. Yes, it is indeed the truth that role of LS in fear has been extensively studied in the previous literature. So to improve the novelty of this manuscript, one must check whether LS play s selective role in retention of tone fear memory or a general role in all types of fear. This point is of vital importance for us to discriminate the exact functions of LS circuits in different forms of fear. Probably, LS GABAergic neurons may equally contribute to contextual fear memory, but through different subsets of neurons or different input-output circuit connections. Regarding my question 3, I think it is nowadays one necessary step to perform optogenetic/chemogenetic manipulations to achieve a more /selective accurate control of neuronal activity than the pharmacological interventions.

Response: Our additional chemogenetic experiments have shown the involvement of GABAergic neurons and dopaminergic neurons in the tone-related fear memory retrieval. Please see our response to your comment 3.

Additional points from referee 1 after having seen our plan to address the referees' comments

Point 1: Thank you for involving me in providing feedback to the authors of this work. To be honest, while I found the responses to the issues 1-3 I raised a bit disappointing, I found the authors' response to the "Cross-comments from referee 1" to be quite satisfactory. Therefore, my opinion is that if the authors follow through with what they propose in their response to "Cross-comments from referee 1," they will adequately address points 1 and 3 raised by me.

Response: Additional experiments have performed as we stated in our plan to address comments. Please see our responses to your cross-comments and your original comment 3.

Point 2: It is also helpful that the authors, as they propose in point 2, specify more clearly in the title and abstract that the work focuses on auditory fear memory retrieval (a term that did not appear before), and similarly in the introduction. In the Introduction, it is also important that they emphasize that previous studies in the literature have already shown that the LS is activated by auditory fear memory retrieval, citing the works (Butler CW et al Learn. Mem 2015; Holschneider DP et al., Neuroimage 2006; Garcia and Jaffard, Eur. J. Neurosci. 1996) that they did not discuss in their response to the issues I raised.

Response: We have now added the phrase "tone-related/auditory fear memory retrieval" in the title, abstract and introduction. We have cited the stated references and discussed the corresponding findings in paragraph 3, page 4.

Dear Dr. Zuo,

Thank you for the submission of your revised manuscript. We have now received the enclosed reports from referee 1 who was asked to assess your full point-by-point response. Referee 1 still has a few more suggestions that I would like you to address and incorporate before we can proceed with the official acceptance of your manuscript. Please submit a point-by-point response to the referee concerns with your final ms.

A few editorial requests will also need to be addressed:

- Please add up to 5 keywords to the ms file.
- Please correct the conflict of interest subheading to "Disclosure and Competing Interests Statement".
- The author credits need to be removed from the ms file. All credits need to be entered during online ms submission.
- The REFERENCE format is not correct: et al should be inserted after the 10th author name. Please use the EMBO reports (Harvard) reference style.
- The FUNDING INFO needs to be part of the Acknowledgments.
- The FIGURE CALLOUTS for the individual panels of Figure 5 are missing; Fig EV8 is called out but there aren't any EV figures, please correct.
- the APPENDIX FILE nomenclature is not correct, it needs the word "Appendix" in the figure titles, legends and ms callouts: it should be Appendix Figure S1, etc. and each legend should follow its figure in the Appendix file. Please correct.
- The Reagents & Tools TABLE needs to be removed from the ms file and uploaded as a separate file.
- "Highlights" need to be removed from the ms.
- Methods and Protocols should be just "Methods".
- Please note that the exact p values are not provided in the legends of figures 2D, F; 3E, 4D, 6F, 7B, 8B, E.
- Our systematic figure analyses of ms to be accepted detected several possible image duplication: Cell/image reuse between Figure 3 B,D and Appendix Figure S2 A,B. And Appendix Figure S1 has reuse within figures D,E & F. A total of 19 images were found to be possibly duplicated. Can you please explain what happened?

I would like to suggest some minor changes to the abstract that needs to be written in present tense. Please let me know whether you agree with the following:

Fear and associated learning and memory are critical for developing defensive behavior. Excessive fear and anxiety are important components of post-traumatic stress disorder. However, the neurobiology of fear conditioning, especially tone-related fear memory retrieval, has not been clearly defined, which limits specific intervention developments for patients with excessive fear and anxiety. Here we show that tone stimuli activate multiple brain regions including the lateral septum (LS). Inhibition of the LS and the connection between basolateral amygdala (BLA) and LS or between LS and ventromedial nucleus of the hypothalamus (VMH) attenuates tone-related fear conditioning and memory retrieval. Inhibiting GABAergic or dopaminergic neurons in the LS also attenuates tone-related fear conditioning. Our data further suggest that fear conditioning is inhibited by blocking orexin B signalling in the LS. Our results indicate that the neural circuitry BLA-LS-VMH is critical for tone-related fear conditioning and memory retrieval, and that GABAergic neurons, dopaminergic neurons and orexin signaling in the LS participate in this auditory fear conditioning.

EMBO press papers are accompanied online by A) a short (1-2 sentences) summary of the findings and their significance, B) 2-3 bullet points highlighting key results and C) a synopsis image that is exactly 550 pixels wide and 200-600 pixels high (the height is variable). The synopsis image should provide a sketch of the major findings, like a graphical abstract. Please note that text needs to be readable at the final size. Please send us this information along with the final manuscript.

Referee #1:

In their revised version, the Authors have addressed many of my comments. However, there are still some aspects that need clarification.

1. In response to my suggestion, the Authors have included throughout the manuscript the fact that they specifically investigated the role of the lateral septum (LS) in auditory memory retrieval, which is good. However, in the Introduction, when presenting the literature, the Authors have made a somewhat peculiar and, above all, incomplete selection. They present studies concerning the involvement of the LS in auditory fear learning but fail to include the two studies (which I previously suggested) that also demonstrate a role of this region in auditory fear memory retrieval. Specifically, Butler CW et al (Learn. Mem 2015) analyzes Fos expression in the LS after auditory fear memory retrieval conducted 4 or 48 hours post-training. Garcia and Jaffard (Eur. J. Neurosci. 1996) show changes in activity in the LS following auditory fear memory retrieval. These two studies should be included and described. Consequently, the statement that there is no data on the involvement of the LS in auditory fear memory retrieval should be nuanced.

2. The Authors find that two pathways are involved in auditory fear memory retrieval. On page 9 of the manuscript, they assert that this may represent a single circuit. There is no data to support this claim. If the Authors wish to test this idea, they should inject an anterograde transsynaptic virus into the BLA and a retrograde virus into the VMH, and then analyze the number of colabeled neurons in the LS. In the absence of such an experiment, this assertion should be removed and proposed only as a hypothesis in the discussion.

3. The Authors report a low level of Fos-activated neurons in the BLA after exposure to the conditioned context or TMT (Fig. 7C-H). This finding is in stark contrast with a vast body of literature that shows increased activity and Fos expression in the BLA following presentation of the conditioned context or TMT. How do the Authors explain this striking discrepancy? Importantly, since it contradicts extensive literature, this data alone cannot strongly support the conclusion that BLA-LH pathway is not involved in context memory retrieval and/or freezing expression. The Authors should present this data with caution and avoid drawing too many conclusions based on it. To clarify this point, it would be useful to repeat the chemogenetic blockade of this pathway immediately before TMT presentation.

Regarding the points raised by Referee 2, the Authors have adequately addressed many of them. However:

Point 1: Fig. S2A. The Authors should also discuss in the manuscript why they observed a large number of TH+ neurons in the LS.

Point 5. The authors propose an explanation for why it is unlikely to observe decreased c-Fos expression in the VMH following chemogenetic inhibition of the BLA→LS→VMH circuitry. However, they should also consider alternative hypotheses, such as the possibility that GABAergic neurons in the LS might target inhibitory neurons in the VMH.

Responses to the referees' comments

We want to thank the editor and referees for their effort in improving our paper. We have revised the manuscript based on the comments. The changes made to address the comments are in red in the manuscript for easy recognition. Our point-by-point responses to the editor's instruction and referees' comments are as follows.

Editor's instructions

Point 1: *Please add up to 5 keywords to the ms file.*

Response: We have deleted one keyword. The total number of keywords is 5 now.

Point 2: *Please correct the conflict of interest subheading to "Disclosure and Competing Interests Statement".*

Response: The subheading is corrected as instructed.

Point 3: *The author credits need to be removed from the ms file. All credits need to be entered during online ms submission.*

Response: The author contributions are now removed.

Point 4: *The REFERENCE format is not correct: et al should be inserted after the 10th author name. Please use the EMBO reports (Harvard) reference style.*

Response: The reference format is now corrected as instructed.

Point 5: *The FUNDING INFO needs to be part of the Acknowledgments.*

Response: The funding information is now moved to the Acknowledgments.

Point 6: *The FIGURE CALLOUTS for the individual panels of Figure 5 are missing; Fig EV8 is called out but there aren't any EV figures, please correct.*

Response: The individual panels of figure 5 are now called out in the second paragraph of page 7. Fig. EV8 is appendix Fig. S8. We have corrected the error.

Point 7: *the APPENDIX FILE nomenclature is not correct, it needs the word "Appendix" in the figure titles, legends and ms callouts: it should be Appendix Figure S1, etc. and each legend should follow its figure in the Appendix file. Please correct.*

Response: We have corrected the incorrect nomenclature in the Appendix file.

Point 8: *The Reagents & Tools TABLE needs to be removed from the ms file and uploaded as a separate file.*

Response: The table is removed from the manuscript file and uploaded as a separate file in the submission website.

Point 9: "Highlights" need to be removed from the ms.

Response: “Highlights” are removed from the manuscript.

Point 10: Methods and Protocols should be just "Methods".

Response: The heading is corrected as instructed.

Point 11: Our systematic figure analyses of ms to be accepted detected several possible image duplication: Cell/image reuse between Figure 3 B,D and Appendix Figure S2 A,B. And Appendix Figure S1 has reuse within figures D,E & F. A total of 19 images were found to be possibly duplicated. Can you please explain what happened?

Response: We often used the same photos for both the main figures and appendix figures when the same brain regions under the same experimental conditions are presented. This situation occurred between Fig. 3 and appendix Fig. S2 and between Fig. 4 and appendix Fig. S3. As instructed in your e-mail, we have stated we use the same photos for the same experimental conditions for those figures. We indeed made some mistakes in placing photos for panels e and f of Appendix Fig. S1. These mistakes are corrected now.

Point 12: I would like to suggest some minor changes to the abstract that needs to be written in present tense. Please let me know whether you agree with the following:

Fear and associated learning and memory are critical for developing defensive behavior. Excessive fear and anxiety are important components of post-traumatic stress disorder. However, the neurobiology of fear conditioning, especially tone-related fear memory retrieval, has not been clearly defined, which limits specific intervention developments for patients with excessive fear and anxiety. Here we show that tone stimuli activate multiple brain regions including the lateral septum (LS). Inhibition of the LS and the connection between basolateral amygdala (BLA) and LS or between LS and ventromedial nucleus of the hypothalamus (VMH) attenuates tone-related fear conditioning and memory retrieval. Inhibiting GABAergic or dopaminergic neurons in the LS also attenuates tone-related fear conditioning. Our data further suggest that fear conditioning is inhibited by blocking orexin B signalling in the LS. Our results indicate that the neural circuitry BLA-LS-VMH is critical for tone-related fear conditioning and memory retrieval, and that GABAergic neurons, dopaminergic neurons and orexin signaling in the LS participate in this auditory fear conditioning.

Response: We have accepted all changes as you suggested except one place where we used “Our data further show that.....” instead of the suggested “Our data further suggest that.....”.

Point 13: EMBO press papers are accompanied online by A) a short (1-2 sentences) summary of the findings and their significance, B) 2-3 bullet points highlighting key results and C) a synopsis image that is exactly 550 pixels wide and 200-600 pixels high (the height is variable). The

synopsis image should provide a sketch of the major findings, like a graphical abstract. Please note that text needs to be readable at the final size. Please send us this information along with the final manuscript.

Response: We have provided items A and B of Synopsis as a separate file. The synopsis image is submitted on-line separately from the items A and B.

Referee 1

In their revised version, the Authors have addressed many of my comments. However, there are still some aspects that need clarification.

Point 1: *In response to my suggestion, the Authors have included throughout the manuscript the fact that they specifically investigated the role of the lateral septum (LS) in auditory memory retrieval, which is good. However, in the Introduction, when presenting the literature, the Authors have made a somewhat peculiar and, above all, incomplete selection. They present studies concerning the involvement of the LS in auditory fear learning but fail to include the two studies (which I previously suggested) that also demonstrate a role of this region in auditory fear memory retrieval. Specifically, Butler CW et al (Learn. Mem 2015) analyzes Fos expression in the LS after auditory fear memory retrieval conducted 4 or 48 hours post-training. Garcia and Jaffard (Eur. J. Neurosci. 1996) show changes in activity in the LS following auditory fear memory retrieval. These two studies should be included and described. Consequently, the statement that there is no data on the involvement of the LS in auditory fear memory retrieval should be nuanced.*

Response: The referred references had been cited and stated in the third paragraph of page 3 in the previous version of the manuscript. We have added the phrase “including auditory fear memory retrieval stimuli” in the sentence to make it clear that the fear stimuli include auditory fear memory retrieval stimuli have been shown to activate lateral septum (LS). However, these previous studies have only shown that the activation of LS by those stimuli and did not perform further experiments to suggest that the activated brain regions are involved in auditory fear memory retrieval. As we have shown in figure 1 that auditory fear memory retrieval stimuli activates many brain regions, we cannot say that all of those activated brain regions are involved in auditory fear memory retrieval.

Point 2: *The Authors find that two pathways are involved in auditory fear memory retrieval. On page 9 of the manuscript, they assert that this may represent a single circuit. There is no data to support this claim. If the Authors wish to test this idea, they should inject an anterograde transsynaptic virus into the BLA and a retrograde virus into the VMH, and then analyze the number of colabeled neurons in the LS. In the absence of such an experiment, this assertion should be removed and proposed only as a hypothesis in the discussion.*

Response: We have replaced the phrase of BLA-LS-VMH circuitry by the phrase of BLA-LS and LS-VMH circuitries throughout the manuscript.

Point 3: *The Authors report a low level of Fos-activated neurons in the BLA after exposure to the conditioned context or TMT (Fig. 7C-H). This finding is in stark contrast with a vast body of*

literature that shows increased activity and Fos expression in the BLA following presentation of the conditioned context or TMT. How do the Authors explain this striking discrepancy? Importantly, since it contradicts extensive literature, this data alone cannot strongly support the conclusion that BLA-LH pathway is not involved in context memory retrieval and/or freezing expression. The Authors should present this data with caution and avoid drawing too many conclusions based on it. To clarify this point, it would be useful to repeat the chemogenetic blockade of this pathway immediately before TMT presentation.

Response: BLA is activated by fear conditioning training stimuli. However, the findings on whether context-related fear memory retrieval stimuli activate BLA are inconsistent. Also, TMT activates LS. However, BLA and VMH are not on the list of brain regions that have increased c-Fos mRNA cells after rodents are exposed to TMT. These results, along with our results that TMT and context-related fear memory retrieval stimuli did not activate BLA and VMH do not provide strong evidence to compel us to perform the chemogenetic experiment to determine the role of BLA and VMH in the TMT-induced freezing behavior. This information is presented in the second paragraph of page 14. We also added the word “may” in the sentence stating the specificity of BLA-LS and LS-VMH pathways in mediating tone-related fear conditioning learning and memory to reduce the tone of certainty.

Regarding the points raised by Referee 2, the Authors have adequately addressed many of them. However:

Point 1: *Fig. S2A. The Authors should also discuss in the manuscript why they observed a large number of TH+ neurons in the LS.*

Response: TH positive cells in the LS of mice have been reported. Transcription factor to facilitate the dopaminergic neuron fate exists in the LS. This information is presented in the first paragraph of page 12.

Point 2: *The authors propose an explanation for why it is unlikely to observe decreased c-Fos expression in the VMH following chemogenetic inhibition of the BLA→LS→VMH circuitry. However, they should also consider alternative hypotheses, such as the possibility that GABAergic neurons in the LS might target inhibitory neurons in the VMH.*

Response: The possibility of targeting inhibitory neurons in the VMH is discussed in the last paragraph of page 13 and the first paragraph of page 14.

Zhiyi Zuo
University of Virginia
Anesthesiology
PO Box 800710
University of Virginia Health System
Charlottesville, VA 22908
United States

Dear Dr. Zuo,

I am very pleased to accept your manuscript for publication in the next available issue of EMBO reports. Thank you for your contribution to our journal.

Yours sincerely,
